# Extracting computational mechanisms from neural data using low-rank RNNs

**Adrian Valente**[*]
École Normale Supérieure
PSL Research University

**Jonathan W. Pillow**
Princeton Neuroscience Institute
Princeton University

**Srdjan Ostojic**
École Normale Supérieure
PSL Research University

## Abstract

An influential framework within systems neuroscience posits that neural computations can be understood in terms of low-dimensional dynamics in recurrent circuits. A number of methods have thus been developed to extract latent dynamical systems from neural recordings, but inferring models that are both predictive and interpretable remains a difficult challenge. Here we propose a new method called Low-rank Inference from Neural Trajectories (LINT), based on a class of low-rank recurrent neural networks (lrRNNs) for which a link between connectivity and dynamics has been previously demonstrated. By fitting such networks to trajectories of neural activity, LINT yields a mechanistic model of latent dynamics, as well as a set of axes for dimensionality reduction and verifiable predictions for inactivations of specific populations of neurons. Here, we first demonstrate the consistency of our method and then apply it to two use cases: (i) we reverse-engineer "black-box" vanilla RNNs trained to perform cognitive tasks, and (ii) we infer latent dynamics and neural contributions from electrophysiological recordings of nonhuman primates performing a similar task.

## 1 Introduction

As large-scale neural recordings in behaving animals are becoming commonplace, a pressing question is how computational principles can be extracted from the electrical activity of thousands of cells. An influential framework posits that neural computations rely on latent low-dimensional dynamics [54, 60] distributed across populations of neurons [66, 43]. In line with this proposal, a number of data-analysis methods have been developed to infer latent dynamics from neural recordings [65, 25, 36, 32, 12, 31, 24, 33, 10, 19, 16, 40, 41, 45, 22] (reviewed in [11]). While these statistical approaches often provide compelling descriptions of the recorded data, they generally lack a mechanistic interpretation that would, for instance, allow them to make predictions for responses to novel interventions on the underlying neural circuits. How to identify mechanistic, predictive models from neural data thus currently remains an important challenge.

Recurrent neural networks (RNNs) have emerged as key models for studying neural computations [64]. Indeed, RNNs can be trained to solve various cognitive tasks, and lead to dynamics surprisingly similar to those observed in neural recordings [29, 56, 5, 61, 39], and have also successfully been trained to reproduce the activity of neurons recorded *in vivo* [38, 6, 14, 35]. While potentially predictive, the obtained networks are however typically challenging to understand mechanistically [55, 28, 27, 26, 63, 57], and ongoing research directions aim at reducing them to simplified, interpretable models, for example through the use of linearized dynamical systems [12, 19, 50] or through "network distillation" methods [44, 22].

---

[*]Correspondence to `adrian.valente@ens.psl.eu`.
Code available at `https://github.com/adrian-valente/lowrank_inference/`

36th Conference on Neural Information Processing Systems (NeurIPS 2022).

Here, we exploit a particular class of interpretable RNNs, namely *low-rank RNNs* (lrRNNs) [30, 3, 9, 46, 47, 58], to develop a new method that extracts mechanistic and predictive low-dimensional models from observed neural activity, and can be applied to both artificial and biological neural networks. Previous work has performed theoretical analyses of low-rank RNNs, either designed or trained on specific tasks [30, 9], but has not introduced methods for fitting them to neural activity. Our method, Low-rank Inference from Neural Trajectories (LINT) infers a minimal rank lrRNN from a dataset and then exploits the theory of low-rank networks to relate the obtained connectivity with low-dimensional dynamics and computations. It produces three outputs: a set of axes for dimensionality reduction of neural population activity, an effective connectivity that implements a latent dynamical system, and predictions for inactivations on specific subsets of recorded neurons (fig. 1a).

Our contributions can be summarized in three steps: first, we verify the consistency of LINT by applying it to data simulated from lrRNNs, and show that it recovers the effective part of the connectivity that reproduces the dynamics and computations. Second, we apply LINT to data generated from full-rank RNNs trained on cognitive tasks, and demonstrate that the resulting lrRNNs can capture their dynamics and offer mechanistic insights into how they function. Notably, we identify a novel population-based mechanism enabling context-dependent switching in a vanilla RNN, and verify it by targeted inactivation experiments. Finally, we apply LINT to electrophysiological recordings in nonhuman primates performing a context-dependent decision making task [29]; this reveals that a rank-1 network can capture most aspects of the dynamics present in the data, and that computations in this neural circuit appear to be supported by a small proportion of all recorded neurons.

## 2 Approach

Here we describe our method for extracting an interpretable low-dimensional projection and dynamical model from neural trajectories by fitting a low-rank recurrent network to data.

**Low-rank RNNs (lrRNNs).** We start from rate-based recurrent neural networks of $N$ units, each characterized by an activation variable $x_i$ which follows the dynamics:

$$\tau \frac{dx_i}{dt} = -x_i + \sum_{j=1}^{N} J_{ij} \phi\left(x_j\right) + \sum_{s=1}^{N_{in}} I_i^{(s)} u_s(t) + \eta_i(t). \tag{1}$$

Here $\boldsymbol{J}$ represents the network connectivity matrix and $\phi$ is a nonlinear transfer function defining the neural firing rate $\phi(x_i)$, which we take here to be $\tanh$. Each network also receives $N_{in}$ input signals $u_s(t)$ via a set of weights $\boldsymbol{I}^{(s)}$ that we refer to as *input vectors*.

Low-rank RNNs represent a subclass of models in which the connectivity matrix is constrained to be of finite rank $R \ll N$ [30, 3, 9]. In this case, $\boldsymbol{J}$ can be written as a sum of outer products of *connectivity vectors* $\boldsymbol{n}^{(r)}$ and $\boldsymbol{m}^{(r)}$:

$$J_{ij} = \frac{1}{N} \sum_{r=1}^{R} m_i^{(r)} n_j^{(r)}. \tag{2}$$

To define a unique representation, we take as $\boldsymbol{m}^{(r)}$ the left singular vectors of the connectivity matrix, and as $\boldsymbol{n}^{(r)}$ the right singular vectors with an appropriate normalization.

**Network inference.** We infer lrRNNs from data by training the connectivity parameters to reproduce recorded neural trajectories (either single-trial or condition-averaged trajectories). Specifically, we use back-propagation on connectivity vector parameters $n_i^{(r)}$ and $m_i^{(r)}$ and input parameters $I_i^{(s)}$ to minimize the squared difference between target and reproduced trajectories:

$$\mathcal{L} = \sum_{c=1}^{C} \sum_{i=1}^{N} \sum_{t=1}^{T} (\phi(x_i^{(c)}(t)) - \phi(\hat{x}_i^{(c)}(t)))^2 \tag{3}$$

where $x_i^{(c)}(t))$ represents the target trajectory in condition $c$, for neuron $i$ and timestep $t$ and $\hat{x}_i^{(c)}(t)$ the corresponding trajectory produced by the model.

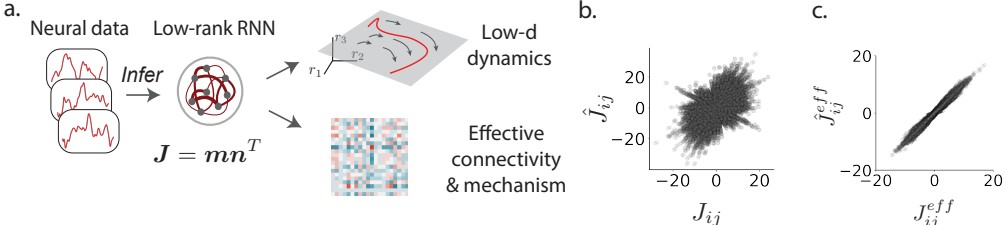

Figure 1: **a**. LINT pipeline: Low-rank RNNs are fitted to either simulated or recorded neural trajectories. The obtained lrRNNs provide interpretable low-dimensional dynamics as well as a computational mechanism based on an effective connectivity structure. **b-c**. LINT tested on trajectories simulated from a rank-1 lrRNN trained to perform the CDM task (see Table 1 for results across tasks). **b**. For each pair of neurons, relationship between the original ($J_{ij}$) and inferred ($\hat{J}_{ij}$) synaptic connectivity ($512^2$ pairs, $r = 0.57$). **c**. For the same networks, relationship between the original effective connectivity ($J_{ij}^{eff}$, see text) and the inferred one ($\hat{J}_{ij}^{eff}$, $r = 0.99$)

**Dimensionality reduction**. One property of lrRNNs is that, in the absence of noise, they constrain the activity vector $\boldsymbol{x}(t)$ to evolve in a low-dimensional subspace spanned by the $R$ connectivity vectors $\boldsymbol{m}^{(r)}$ and the $N_{in}$ input vectors $\boldsymbol{I}^{(s)}$ [3]. The trial-averaged trajectories therefore explore at most $R + N_{in}$ dimensions and can be parametrized as:

$$\boldsymbol{x}(t) = \sum_{r=1}^{R} \kappa_r(t)\boldsymbol{m}^{(r)} + \sum_{s=1}^{N_{in}} v_s(t)\boldsymbol{I}^{(s)}, \tag{4}$$

where $v_s(t)$ are the low-pass filtered input signals $u_s(t)$ (see Appendix A), and $\kappa_r$ are a set of latent variables generated by recurrent activity. Thus, lrRNNs provide by design a reduction of $N$-dimensional neural activity to at most $R + N_{in}$ dimensions that can be directly interpreted in terms of components on a *recurrent subspace* spanned by the connectivity vectors $\boldsymbol{m}^{(r)}$ and on an *input subspace* spanned by inputs vectors $\boldsymbol{I}^{(s)}$ [61].

**Latent dynamics**. In lrRNNs, the latent variables $\kappa_r(t)$ form a non-linear low-dimensional dynamical system, described by:

$$\frac{d}{dt}\boldsymbol{\kappa}(t) = F(\boldsymbol{\kappa}(t), \boldsymbol{u}(t)) \tag{5}$$

where $\boldsymbol{\kappa}(t) = \{\kappa_r(t)\}_{r=1..R}$ is an $R$-dimensional vector representing the activity on the recurrent subspace, $\boldsymbol{u}(t)$ represents the input signals, and $F$ is a non-linear function that can be directly determined from network connectivity parameters [3, 9] (Appendix A). Moreover, it can be shown that rank-$R$ networks are universal approximators of $R$-dimensional dynamical systems [3], and thus that the function $F$ can approximate any non-linear mapping in $R$ dimensions.

**Task-optimized RNNs**. We applied our method first to trajectories produced by task-optimized RNNs, which produce an output signal $z(t)$ from the recurrent dynamics (1) via a linear readout:

$$z(t) = \sum_{i=1}^{N} w_i \phi(x_i(t)) \tag{6}$$

In task-optimized networks, the parameters are trained with backpropagation through time to minimize the squared error between the output $z(t)$ and a target $z^*(t)$. In this work, we consider four cognitive tasks: Decision Making (DM), Working Memory (WM), Context-Dependent Decision Making (CDM, fig. 2a) and Delayed Match-to-Sample (DMS) (see Appendix B for task definitions). To analyze the computations underlying these tasks, we generated neural trajectories from both low-rank (in section 3.1) and full-rank (see section 3.2) task-optimized RNNs. All networks were implemented in pytorch [34]; training details can be found in Appendix C.

Table 1: Synthetic data validation results for LINT
(CC: connectivity correlation, ECC: effective connectivity correlation)

| Task | Rank | Trajectory $R^2$ | CC | ECC |
|---|---|---|---|---|
| Decision Making (DM) | 1 | 0.97 | 0.50 | 0.99 |
| Working Memory | 2 | 0.96 | 0.43 | 0.93 |
| Context-dependent DM | 1 | 0.91 | 0.57 | 0.99 |
| Delayed Match-to-Sample | 2 | 0.98 | 0.39 | 0.63 |

## 3 Results

### 3.1 Validation with synthetic data and effective aspects of connectivity

We first validated the capacity of the LINT method to recover low-rank connectivity features and reproduce neural trajectories on data simulated from lrRNNs trained on cognitive tasks. For this, we first trained lrRNNs with 512 neurons to produce a correct behavioral output on four systems neuroscience tasks, each time retaining the minimal rank solution [9] (see Table 1). For each task-optimized lrRNN, we then generated simulated trajectories corresponding to trials in the trained task, and applied our method to fit these trajectories using equivalent models with the same rank and number of neurons. We found that the inferred networks were able to reliably reproduce the original trajectories when fed with the same inputs, as quantified by the $R^2$ scores between original and fitted trajectories. Although the inferred networks were not explicitly trained on behavioral outputs, they were able to perform their task accurately when plugged to the original readout vector, implying that they also captured behavioral aspects of the original networks.

The output of LINT is an inferred low-rank connectivity that we compared with the original one. We noted that the inferred connectivity weights are not identical to the original ones, although a certain degree of correlation is present (fig. 1b and Table 1, column CC for connectivity correlation). However, from a theoretical point of view, a range of different connectivity matrices can lead to identical dynamics [37, 13, 3]. The low-rank framework allows for a precise characterization of the relevant part of the connectivity, and in particular allows us to define an effective connectivity matrix $\boldsymbol{J}^{\text{eff}}$ that captures the minimal features required to obtain certain dynamics. Indeed, for this class of networks, the low-dimensional dynamics described by equations (4)-(5) depend on the exact entries on the $\boldsymbol{m}^{(r)}$ and $\boldsymbol{I}^{(s)}$ vectors, but not on the individual entries on the $\boldsymbol{n}^{(r)}$ vectors. Instead, $\boldsymbol{n}^{(r)}$ vectors influence the dynamics only through their global projections on the $\boldsymbol{m}^{(r)}$ and $\boldsymbol{I}^{(s)}$ vectors (see details in Appendix A). Thus, the components of $\boldsymbol{n}^{(r)}$ orthogonal to the space spanned by $\boldsymbol{m}^{(r)}$ and $\boldsymbol{I}^{(s)}$ are irrelevant for the dynamics, and removing them leads to an effective connectivity matrix $\boldsymbol{J}^{\text{eff}}$ which captures the minimal features required for obtaining particular dynamics. Our results on the synthetic data showed indeed a very high degree of similarity between original and inferred effective connectivities, demonstrating that LINT recovered the relevant aspects of the neural connectivity (Table 1, ECC for Effective Connectivity Correlation, and fig. 1c).

In conclusion, LINT is able to accurately fit neural trajectories generated by low-rank RNNs, to infer networks that give rise to similar trajectories and behavioral performance, and to recover the essential features of the connectivity.

### 3.2 Application to reverse-engineering full-rank RNNs

We next ask to which extent our method can infer computational mechanisms from activity generated by unconstrained, full-rank networks. Indeed, RNNs trained on simple tasks without a rank constraint often exhibit low-dimensional dynamics that can be related to the way computations unfold in the network [55, 28, 27, 26, 57]. It has been found that such low-dimensional dynamics can be reproduced by low-rank networks [3], yet the best approach for inferring the corresponding low-rank connectivity remains to be determined. Here we show that our method vastly outperforms a direct low-rank approximation of the connectivity matrix based on truncating the SVD [47]. We then demonstrate how the inferred low-rank models can be used to interpret the computational mechanisms in the original full-rank networks.

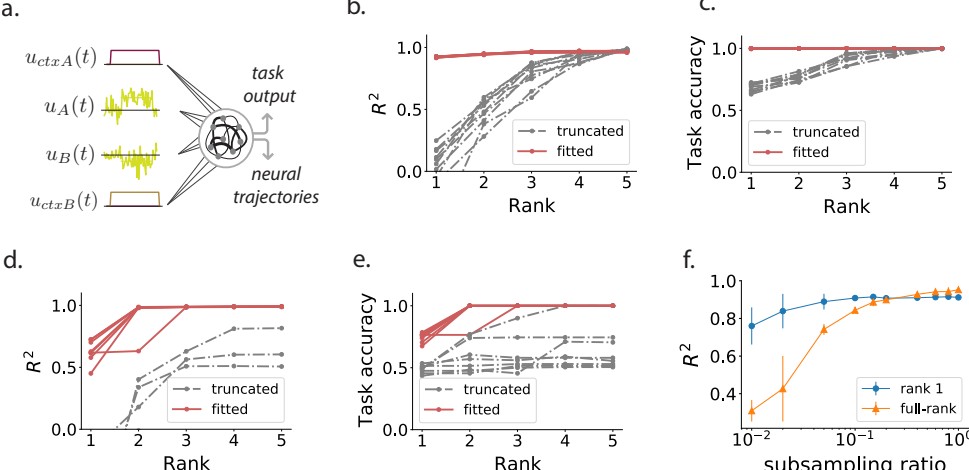

Figure 2: LINT applied to full-rank task-optimized RNNs. **a**. Description of the context-dependent decision making (CDM) task inputs and outputs. Neural trajectories correspond to the neural firing rates, task output refers to a linear readout as in eq. 6. **b**. For the CDM task, similarity between trajectories produced by original full-rank networks and a series of fitted networks of increasing rank (red), or low-rank networks obtained by truncating the original connectivity matrix (grey). Each line corresponds to a randomly initialized full-rank network. **c**. Task accuracy of the same networks when associated with the original task readout. **d-e**. Same as b-c. for the delayed match-to-sample (DMS) task. **f**. For the CDM task, similarities between trajectories of the original and fitted networks of rank one (blue) or full-rank (orange), when networks are fitted only to a random subset of neurons of the original network. Error bars: mean ±1std over 10 random subsamples for each ratio value (see also sup. fig. 8).

### 3.2.1 Extracting low-dimensional dynamics through low-rank connectivity

We considered full-rank, vanilla RNNs trained on two systems neuroscience tasks, respectively context-dependent decision-making[29] (CDM) and delayed match-to-sample[5] (DMS). The full-rank RNNs reached a 100% accuracy on each task, and—as expected—exhibited low-dimensional dynamics. From these RNNs, we generated trajectories corresponding to trials in the trained tasks, and then used LINT to infer lrRNNs of increasing rank.

In the CDM task, a network received two signal inputs which varied randomly around a positive or negative mean, as well as two binary context cues, one of which was set to 1 in each trial. The network was trained to output the average sign of the signal indicated by the active context, while ignoring the other signal (fig. 2a, see Appendix B for task details). Applying LINT, we found that rank-1 networks are sufficient to accurately reproduce the trajectories of the trained full-rank networks. Increasing the rank improved the goodness-of-fit only marginally (fig. 2b). In contrast, low-rank networks obtained by truncating the SVD of the trained full-rank connectivity matrix required a rank of 4 or 5 to reach a comparable accuracy (fig. 2b), and much higher for different hyperparameters (sup. fig. 1). This shows that LINT captures a simplified connectivity structure that could not be trivially extracted from the original connectivity. Moreover, even though not explicitly trained on the behavioral outputs, the inferred rank-1 networks performed the task correctly when plugged onto the original readout, implying that they faithfully captured the task-related dynamics (fig. 2c).

For the DMS task, we similarly found that rank-2 networks were able both to capture the dynamics of the trained full-rank network and to accurately perform the task, with no notable improvement when the rank is increased (fig. 2d-e). As for the CDM task, LINT vastly outperformed direct truncation of the full-rank connectivity matrix. For both these tasks, reproducing the experiment over a larger number of full-rank RNNs, trained with diverse hyperparameters and random seeds led to similar results (sup. fig. 1).

Finally, it is important to note that the CDM and DMS tasks are particularly prone to low-dimensional implementations, which is not necessarily the case for all computations of interest in systems neuroscience [18]. To probe the capacities of LINT with higher-dimensional tasks, we applied it to full-rank networks trained on the K-back task, where a network receives a random number $P$ of stimulations in $\{-1, 1\}$ and has to output the $K$-th last stimulation received, a task directly inspired from sequence working memory paradigms [62]. This task requires the implementation of an internal working memory with a capacity of at least $K$ bits, and should thus require a rank at least $K$. We indeed observed that the rank identified by LINT increased with $K$, but only linearly (see details in appendix F and sup. fig. 9). These results show that LINT is also helpful in probing the dimensionality required for more complex tasks.

**Subsampling**. In experimental settings, one cannot expect to have access to all units of a network. It is thus important to assess whether we can still recover the relevant low-dimensional dynamics and information when only a handful of neurons from a network are recorded. We considered a full-rank network trained on the CDM task, and fitted networks to its trajectories, either without constraining their rank or by fixing it to one. Without subsampling, the inferred full-rank networks slightly outperformed rank-1 networks in reproducing the original trajectories. Yet when considering subsamples of neurons, rank-1 networks appear to be more robust than unconstrained ones, maintaining a good performance until subsampling ratios as low as 1% of original neurons (fig. 2f). This experiment was reproduced on the DMS task setting, showing a similar advantage for low-rank fitted networks (sup. fig. 8).

### 3.2.2 Extracting computational mechanisms from inferred low-rank connectivity

Low-rank models of behavioral tasks open the door to mechanistic interpretations of the underlying dynamics [30, 9]. Here we show how LINT allows us to extract computational mechanisms from full-rank networks performing the CDM task.

A first output of LINT is a set of interpretable axes defining a task-related subspace for dimensionality reduction. The inferred rank-1 model of the CDM task specifically yields five axes that correspond to the two stimulus inputs, the two context inputs and an internal latent variable, generated by recurrent connectivity, which represents integrated evidence and therefore choice. Projecting the activity of the full-rank network along these axes shows how low-dimensional dynamics transform inputs into the choice output (fig. 3). In this case, the axes determined by LINT are closely related to those obtained by standard targeted dimensionality reduction (TDR, sup. fig. 2) [29]. However, in contrast with TDR, the axes inferred by LINT correspond to connectivity features causing the low-dimensional dynamics, and provide a method for an unsupervised discovery of recurrently-generated latent variables. In particular, we did not *a priori* specify that the activity along $m$ should encode choice, but directly observed that choice was generated along that axis through recurrent dynamics.

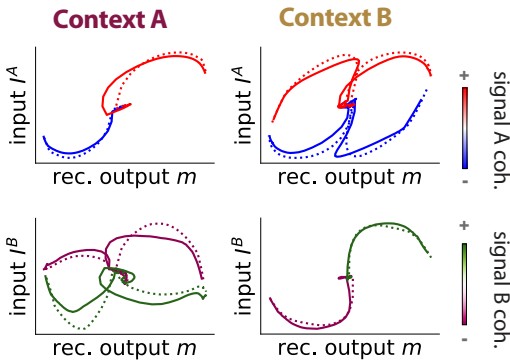

Figure 3: Low-dimensional projections produced by LINT. Trial-averaged trajectories for different task conditions, in the original full-rank network (full lines) and the inferred low-rank network (dashed lines), projected on axes obtained from the fitted lrRNN connectivity: input A and B axes, and the output axis of the rank-1 recurrent connectivity $m$, which encodes choice. All trajectories start at center.

Going beyond dimensionality reduction, LINT extracts an effective low-dimensional model of the task. For the CDM task, the inferred model is a one-dimensional, non-linear latent dynamical system, where the latent variable integrates the relevant evidence. To understand how context-dependent selection of the evidence was performed, we analyzed the connectivity parameters of the inferred rank-1 RNN. Indeed, recent work has introduced a clustering method for analyzing low-rank connectivity, and has showed that in trained lrRNNs, context-dependent integration relies on gain-modulation mechanism mediated by two distinct populations of neurons [9]. Applying a similar clustering approach to rank-1 networks inferred by LINT led to a

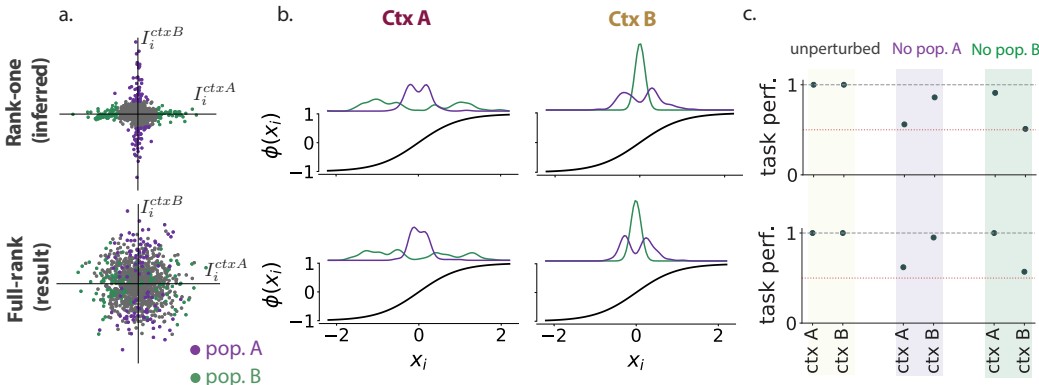

Figure 4: Extracting mechanisms and testing predictions from a low-rank network (top) inferred from a full-rank network trained on the CDM task (bottom). **a**. Three populations of neurons (grey, green and purple) are identified by clustering connectivity parameters in the fitted rank-1 network (top). Here, we plot the same populations in joint distributions of contextual input vectors in the rank-1 (top) vs full-rank (bottom) networks. The populations are not directly identifiable from input parameters in the original network (bottom). **b**. Neural transfer function (tanh) and smoothed distributions of mean neural activity for populations A and B in the inferred rank-1 network (top) and the full-rank network (bottom). **c**. Inactivation experiments: task performance on each context for unperturbed networks and after inactivating populations A and B in the fitted rank-1 network (top) and inactivating the same neurons in the original unconstrained network (bottom). Red dots: chance level.

comparable mechanism. Specifically, this method identified two populations distinguished by strong values of the fitted weights for the context-cue inputs (fig. 4a top and sup. fig. 2). In each context, these context-cue inputs place the neurons belonging to the two populations at different positions on the non-linear transform function $\phi(x)$ (fig. 4b top), thereby modulating in opposite ways their gains (defined for each neuron as the local slope $\phi'(x)$). Combined with different statistics of the connectivity parameters on each of the populations, this modulation is sufficient to implement the desired context-switching behavior of the network (see details in Appendix D). The same pattern of gain modulation was found in the original full-rank network, indicating that it is using a similar mechanism (fig. 4b bottom).

The computational mechanism extracted using LINT produces predictions for inactivations that can be directly tested in the original full-rank model. The analysis described above assigns neurons to different populations that have specific computational roles. Specifically, in the inferred rank-1 network, inactivating separately each population leads to a specific loss of performance in a single context, but not the other one (fig. 4c top and sup. fig. 2 for detailed error patterns). Since individual neurons in the original networks directly correspond to neurons in the rank-1 network on a one-to-one basis, the identified populations can be directly mapped back onto the original network to reproduce the inactivation experiment. The predicted effects of inactivations on context-dependent performance were directly replicated in the full-rank network, demonstrating that it relies on the identified computational mechanism (fig. 4c bottom and sup. fig. 2). One can note that although the two identified populations clearly stand out in the inferred rank-1 connectivity, they are not directly apparent in the original full-rank connectivity (fig. 4a bottom), showing that although the mechanism extracted by LINT applies to the full-rank network, using an lrRNN was a necessary step to uncover it.

## 3.3 Application to neural recordings

We next applied our method to *in vivo* recordings. We considered here electrophysiological recordings from non-human primates performing a context-dependent decision-making task similar to that studied in previous sections [29]. More specifically, two macaques (designed by A and F) were presented with random dots stimuli that varied along two dimensions: the overall motion direction of the dots, and their overall color, ranging from two extremes with a set of intermediary coherences in between. At the beginning of each trial, a cue indicated a context for the trial, ordering the

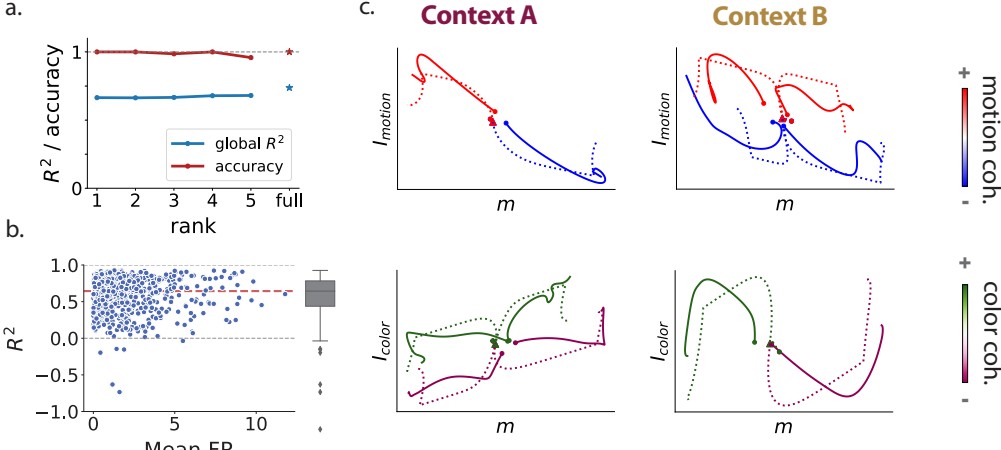

Figure 5: LINT applied to neural recordings from a non-human primate performing a context-dependent task (monkey A) [29]. **a**. Global $R^2$ of model trajectories with respect to original ones, and task accuracy of inferred networks of increasing ranks. **b**. For the rank-1 inferred network, $R^2$ of each recorded neuron plotted against its average data firing rate, median $R^2$ in red, with marginal box-plot of $R^2$ values. 14 neurons with an $R^2 < -1$ are not shown, all with a mean firing rate $< 2.2$ (see sup. fig. 4). **c**. Trial-averaged trajectories in the recorded data (full lines) and for the rank-1 model (dashed lines), for different task conditions (only one coherence value for each coherence sign is plotted here), projected on axes identified by LINT (motion and color input axes and the output axis $m$ of the rank-1 recurrent matrix). Trajectories start at the center.

subject to report either the average motion or color with an eye saccade, while ignoring the irrelevant evidence. Non-simultaneous recordings were performed in the frontal eye field (FEF) an area of the macaque prefrontal cortex, with respectively 727 neurons for monkey A and 574 neurons for monkey F recorded in all 72 conditions the task presented (ignoring error trials).

For the analysis, we started by binning (5 ms) and smoothing (50 ms std Gaussian window) spike train data, and then computed the trial-averaged response of every neuron in each of the task conditions, forming a pseudo-population tensor of size $N \times T \times 72$ with $N$ the number of neurons in each monkey and $T = 150$ the number of discrete time steps. We denoised this tensor by projecting its first axis on the subspace spanned by the top 12 principal components of the pseudo-population activity, and then projecting back to the high-dimensional space spanned by all neurons [29]. Finally, in order to consider only task-related neural activity, we subtracted from each neuron's trajectory its condition-averaged mean. We denote the entries of the final obtained tensor by $\tilde{x}_{i,t,c}$.

We applied LINT to the obtained trajectories, training the activation of each unit in the low-rank RNN (noted $x_{i,t,c}$) to match the corresponding pre-processed activity. Importantly, the inferred networks received inputs following the same structure than in the CDM task of previous sections, that is two noisy signal inputs and two contextual cue inputs (fig. 2a), and were left unconstrained for the first 350 ms of each trial while receiving only contextual cues, to account for the original task procedure. In particular, we made the implicit hypothesis that choice signals are generated by the recurrent activity of the network from received inputs. The quality of fits was quantified by leaving out a random subset of 8 conditions during network inference, and evaluating the $R^2$ of fitted networks on these left-out conditions.

We found that for both monkeys the neural activity was well reproduced by a rank-1 network, with minimal improvements when the rank was increased (fig. 5a and sup. fig. 5, $R^2 = 0.66$ for monkey A, and $R^2 = 0.57$ for monkey F). Moreover, when simulated independently, the inferred rank-1 networks were able to perform the task by adding a linear readout (accuracy curves in fig. 5a, sup. fig. 5). This was the case even though the recurrent and input connections were not trained on task performance, demonstrating that the reproduced trajectories contained information about the choice made by the monkey. The activity of both original and reproduced trajectories can moreover be projected on the axes identified by LINT, providing a geometrical picture of task execution (fig. 5c). These axes were

closely related to those found by targeted dimensionality reduction: notably the context, motion, and color axes identified by TDR closely match the corresponding input axes in the inferred network, while the choice TDR axis could be identified to the $m$ axis of the rank-1 recurrent connectivity (sup. figs. 4,5). The projections of the original activity on the connectivity axes (full lines on fig. 5b) therefore showed how inferred connectivity explains the geometry of recorded data.

A closer look at the distribution of inferred connectivity weights provided more information on the neural mechanisms at play (sup. figs. 6, 7). In contrast to the fits of task-optimized networks, we observed for both monkeys a clearly non-normal, heavy-tailed distribution of inferred connectivity parameters (Fisher's kurtosis between 4.9 and 62.7 for different connectivity parameters), echoing some past observations on biological synaptic weights [51, 4]. Separating with a clustering algorithm (GMM) large and small-weight neurons showed that the latter could be inactivated without affecting the task performance of the network in a significant way (sup. figs. 6, 7), even though they represented a majority of neurons in the circuits (570/727 neurons in monkey A, 389/574 in monkey F). This suggests that a minority of neurons with large connectivity weights supports the computation performed by these networks.

# 4   Discussion

In this paper, we introduced LINT, a new method for inferring a latent dynamical system from neural activity based on the theory of low-rank RNNs. This method yields a mechanistic, interpretable and predictive model of neural trajectories, and bridges different levels of analysis, from state-space geometry to neural connectivity. After verifying the consistency of our method, we demonstrated its potential to extract the mechanisms used by "black-box" vanilla RNNs trained to perform cognitive tasks. In particular, we found that low-rank RNNs reproduced well the dynamics of full-rank RNNs trained on a variety of tasks and across a range of hyperparameters. Moreover, the obtained lrRNNs provided interpretable low-dimensional projections of the fitted activity, as well as predictions for the effects of inactivations of specific populations of neurons. These predictions were verified in the original network, validating computational mechanisms derived from inferred low-rank connectivity weights. Finally, we applied LINT to neural activity recorded in the prefrontal cortex of nonhuman primates during a context-dependent decision-making task. LINT was able to infer rank-1 networks that reproduced both neural activity and task performance, from which low-dimensional projections and interpretable connectivity could be extracted.

Our results pave the way for a better understanding of how RNNs perform computations. In particular, LINT is complementary to other approaches seeking to reverse-engineer RNNs [55, 29, 50] in that it infers an effective connectivity that links low-dimensional dynamics with interactions at the single neuron level, opening possibilities for goal-oriented interventions and a better control on the behavior of trained networks. Moreover, LINT leads to compressed, distilled versions of large networks that can be of practical interest to engineers, as suggested by recent research in low-rank training of deep neural networks [20]. Extensions to reverse-engineering more complex architectures like GRUs or LSTMs, which have already been shown to rely on low-dimensional trajectories [28, 26, 21] would be an interesting direction for future research.

It is worth noting that other statistical methods such as mTDR [1] can provide a better fit to the same data, but do not identify a dynamical system that generates the trajectories. LINT is related to switching latent dynamical systems [10, 50, 24, 19, 12, 58] in that it bridges descriptive latent-space analyses like mTDR with methods that identify a predictive model of the dynamics, such as LFADS [33]. A distinctive feature of LINT is however that (unlike LFADS) it retains a mechanistic description at the level of neural connectivity of how the dynamics emerge from interactions between neurons.

As a number of other data analysis methods for neuroscience, our approach relies on the ubiquitous observation that neural dynamics appear to be confined to low-dimensional manifolds within the activity state-space [7, 17, 43]. Recent works have proposed that this low-dimensionality might be an artifact stemming from the simplicity of the examined tasks [18], or have reported higher-dimensional activity patterns [53, 23, 59]. This raises the question of how far our method could account for activity in more complex tasks. Our results on the K-back task suggest that LINT scales well with task dimensionality, and hint to a possible compositional implementation of internal computations

that could be unpacked by breaking them into simpler primitives [63, 8]. Further confronting our and related methods to higher-dimensional task designs however remains an important endeavour.

Distinguishing signals generated by dynamics within a given area from external inputs it receives is a major challenge for any interpretation of neural recordings, and a focus of recent work [16, 45, 42]. Most methods for inferring latent dynamics from activity data rely on *ad hoc* assumptions about the inputs received by a circuit. Following previous work [29], when analyzing recordings from the prefrontal cortex we hypothesized that choice was recurrently generated within the recorded area. LINT however potentially provides a new approach for testing hypotheses on input structure, by comparing models fits obtained from single-trial recordings, or by building multi-area models [2] from large-scale recordings that are increasingly available nowadays [49, 48, 52, 35].

## Acknowledgments and Disclosure of Funding

The authors would like to thank Valerio Mante for kindly providing access to the monkey data, and João Barbosa for relevant discussions. AV and SO were supported by the NIH Brain Initiative project U01-NS122123 and the program "Ecoles Universitaires de Recherche" launched by the French Government and implemented by the ANR, with the reference ANR-17-EURE-0017. JWP was supported by grants from the Simons Collaboration on the Global Brain (SCGB AWD543027), the NIH BRAIN initiative (R01EB026946), and a U19 NIH-NINDS BRAIN Initiative Award (5U19NS104648).

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
