# A Low-rank RNN dynamics

Here, we detail the theoretical link between connectivity and low-dimensional dynamics in low-rank RNNs, and outline its consequences for the main results. This section mostly summarizes the mathematical results of the papers [30], [3] and [9]. Sections A.3 and A.4 can be read without the preceding ones.

## A.1 Low-dimensional dynamics

Let us first recall the formalism. We consider a network of $N$ units, each characterized by an activation that follows the dynamics given in eq. (1) of the main text:

$$\tau \frac{dx_i}{dt} = -x_i + \sum_{j=1}^{N} J_{ij} \phi\left(x_j\right) + \sum_{s=1}^{N_{in}} I_i^{(s)} u_s(t) + \eta_i(t). \tag{7}$$

with a connectivity matrix of rank $R \ll N$, written as:

$$J_{ij} = \frac{1}{N} \sum_{r=1}^{R} m_i^{(r)} n_j^{(r)}. \tag{8}$$

where, to remove all degeneracy, we define vectors $\boldsymbol{m}^{(r)}$ as the left singular vectors of the connectivity matrix, and vectors $\boldsymbol{n}^{(r)}$ as the right singular vectors multiplied by the corresponding singular value (determined up to a change in sign). A consequence of this definition is that the $\boldsymbol{m}^{(r)}$ vectors are all orthogonal to each other, and the $\boldsymbol{n}^{(r)}$ vectors as well. As a further simplifying assumption we will also assume that the $\boldsymbol{I}^{(s)}$ vectors are also orthogonal to each other, and to each of the $\boldsymbol{m}^{(r)}$ vectors.

Injecting equation (8) into the differential equation (7) gives the vector dynamics:

$$\tau \frac{d\boldsymbol{x}}{dt} = -\boldsymbol{x}(t) + \sum_{r=1}^{R} \boldsymbol{m}^{(r)} \boldsymbol{n}^{(r)\top} \phi(\boldsymbol{x}(t)) + \sum_{s=1}^{N_{in}} \boldsymbol{I}^{(s)} u_s(t) + \boldsymbol{\eta}(t), \tag{9}$$

where $\phi$ is applied in an elementwise manner. A direct consequence of (9) is is that the vector of neural activations $\boldsymbol{x}(t)$ is constrained to evolve in a subspace spanned by the vectors $\boldsymbol{m}^{(r)}$ and $\boldsymbol{I}^{(s)}$, of dimension $R + N_{in}$. Since these vectors form a basis of this subspace, one can decompose $\boldsymbol{x}(t)$ as:

$$\boldsymbol{x}(t) = \sum_{r=1}^{R} \kappa_r(t) \boldsymbol{m}^{(r)} + \sum_{s=1}^{N_{in}} v_s(t) \boldsymbol{I}^{(s)}, \tag{10}$$

where the $\kappa_r(t)$ are a set of recurrently generated latent variables, defined as the projection of $\boldsymbol{x}(t)$ on $\boldsymbol{m}^{(r)}$:

$$\kappa_r(t) := \frac{1}{\|\boldsymbol{m}^{(r)}\|^2} \sum_{j=1}^{N} m_j^{(r)} x_j(t), \tag{11}$$

and the $v_s(t)$ are low-pass filtered versions of the input signals $u_s(t)$, following:

$$\tau \frac{dv_s}{dt} = -v_s(t) + u_s(t). \tag{12}$$

The dynamics of the internal variables $\kappa_r(t)$ can also be described by a set of self-consistent differential equations, obtained by projecting equation (9) on each $\boldsymbol{m}^{(r)}$:

$$\tau \frac{d\kappa_r}{dt} = -\kappa_r(t) + \underbrace{\frac{1}{N} \sum_{j=1}^{N} n_j^{(r)} \phi(x_j(t))}_{:=\kappa_r^{rec}(t)}, \tag{13}$$

(where we have taken into account the orthogonality of each input vector $\boldsymbol{I}^{(s)}$ with the vectors $\boldsymbol{m}^{(r)}$ as a simplifying assumption). The second term of the r.h.s. of this equation, $\kappa_r^{rec}(t)$, which represents

the recurrent contribution to the input of $r$-th latent variable, can be expanded as:

$$\kappa_r^{rec}(t) = \frac{1}{N} \sum_{j=1}^{N} n_j^{(r)} \phi \left( \sum_{r'=1}^{R} \kappa_r'(t) m_j^{(r')} + \sum_{s=1}^{N_{in}} I_j^s v_s(t) \right), \tag{14}$$

which only depends on the set of $\kappa_r(t)$ and $v_s(t)$ functions, closing the system of equations.

Equations (13)-(14) show that a rank-$R$ RNN is an $R$-dimensional dynamical system, which can be written as:

$$\frac{d}{dt} \boldsymbol{\kappa}(t) = F(\boldsymbol{\kappa}(t), \boldsymbol{u}(t)) \tag{15}$$

with $\boldsymbol{\kappa}(t)$ an $R$-dimensional state-space vector, and $\boldsymbol{u}(t)$ the $N_{in}$-dimensional vector of input signals. Here, $F$ represents a non-linear map from $\mathbb{R}^{N_{in}+R}$ to $\mathbb{R}^R$ that depends exclusively on the parameters defining the network connectivity, that is the $\{I_i^{(s)}, m_i^{(r)}, n_i^{(r)}\}_{s,r,i}$. It can be shown that these parameters can be chosen in order to approximate any non-linear map $F$, so that the class of rank-$R$ RNN are universal approximators for $R$-dimensional dynamical systems [3].

## A.2 Mean-field approximation

Moreover, the map $F$ can be more explicitly defined in the large network limit $N \to \infty$, if the connectivity parameters follow a particular distribution, namely a joint mixture-of-Gaussians distribution [3, 9]. Notably, in this framework, each neuron is characterized by its weights on the $N_{in}$ input vectors and on the $2R$ recurrent connectivity vectors. Hence every neuron $i$ can be seen as a point in a $(N_{in} + 2R)$-dimensional *connectivity space* of coordinates $(I_i^{(1)}, \ldots, I_i^{(N_{in})}, n_i^{(1)}, \ldots, n_i^{(R)}, m_i^{(1)}, \ldots, m_i^{(R)}) := (\underline{I}, \underline{n}, \underline{m})$.

We can then assume that these points are sampled from a probability distribution :

$$P(I^{(1)}, \ldots, I^{(N_{in})}, n^{(1)}, \ldots, n^{(R)}, m^{(1)}, \ldots, m^{(R)}) := P(\underline{I}, \underline{n}, \underline{m}). \tag{16}$$

We will take the assumption that the distribution $P(\cdot)$ is a mixture-of-Gaussians as in [3, 9]. It can then be written as:

$$P(\underline{n}, \underline{m}, \underline{I}, \underline{w}) := \sum_{p=1}^{P} \alpha_p \mathcal{N}(\boldsymbol{\mu}_p, \boldsymbol{\Sigma}_p), \tag{17}$$

with $p$ components to the mixture, each associated to the probability $\alpha_p$, the mean $\boldsymbol{\mu}_p$ and the covariance matrix $\boldsymbol{\Sigma}_p$. We will add the simplifying assumption that all components are zero-centered, and vary just in terms of their covariances, so that $\boldsymbol{\mu}_p = \mathbf{0}$. Then, following a mean-field approximation, the dynamics in equation (13) can be simplified for each $r$ to:

$$\frac{d\kappa_r}{dt} = -\kappa_r(t) + \sum_{r'=1}^{R} \tilde{\sigma}_{n^{(r)} m^{(r')}} \kappa_{r'}(t) + \sum_{s=1}^{N_{in}} \tilde{\sigma}_{n^{(r)} I^{(s)}} v_s(t). \tag{18}$$

where for any connectivity parameters $a, b$ in $\{m^{(r)}, n^{(r)}, I^{(s)}\}$, $\tilde{\sigma}_{ab}$ represents an *effective coupling*, given by a weighted average over populations of the associated covariances:

$$\tilde{\sigma}_{ab} = \sum_{p=1}^{P} \alpha_p \sigma_{ab}^{(p)} \langle \Phi' \rangle_p, \tag{19}$$

where $\sigma_{ab}^{(p)}$ represent the covariance between parameters $a$ and $b$ in the $p$-th component (so that e.g. $\sigma_{n^{(1)} m^{(2)}}^{(p)} = \frac{1}{N} \sum_{i \in p} n_i^{(1)} m_i^{(2)}$), and $\langle \Phi' \rangle_p$ represents an average gain over neurons of the $p$-th component, defined as the average over these neurons of the derivative of the neural transfer

function $\phi'(x_i)$. Since these gains are distributed normally on each population, this average depends non-linearly on the $\kappa_r$ and $v_s$ values following equation:

$$\langle\Phi'\rangle_p = \langle\Phi'\rangle\left(\sqrt{\sum_{r'=1}^{R}(\sigma_{m^{(r')}}^{(p)})^2\kappa_{r'}^2 + \sum_{s=1}^{N_{in}}(\sigma_{I^{(s)}}^{(p)})^2 v_s^2}\right), \tag{20}$$

where $\langle\Phi'\rangle(\Delta)$ is the Gaussian integral:

$$\langle\Phi'\rangle(\Delta) = \frac{1}{\sqrt{2\pi}}\int_{-\infty}^{+\infty}dz\,e^{-z^2/2}\phi'(\Delta z). \tag{21}$$

These equations finish to make the system given by equations (18) a closed system of non-linear coupled differential equations where the unknowns are the variables $\kappa_r(t)$.

### A.3 Intuitive interpretation of the mean-field results

A more intuitive interpretation of those derivations is that equation (18) locally approximates the recurrent dynamics as a linear dynamical system (LDS), driven by the effective couplings between latent variables $\tilde{\sigma}_{ab}$ (which can vary depending on the state of the network $\boldsymbol{\kappa}(t)$ and on the inputs $\boldsymbol{u}(t)$). More specifically, the effective coupling from latent variable $\kappa_r$ to latent variable $\kappa_{r'}$ (where possibly $r' = r$) is given by $\sigma_{n^{(r')}m^{r)}}$, and the effective coupling from input $s$ to latent variable $r$ by $\sigma_{n^{(r)}I^{(s)}}$.

These effective couplings can be modulated through equation (20) by inputs as well as the latent state of the network, leading to different dynamics in different parts of the neural state-space. In particular, inputs can have a purely modulating effect by modifying only the gain of specific population, hence modifying the effective couplings and the resulting network dynamics.

### A.4 Notion of effective connectivity

A consequence of equation (18) is notably is that the low-dimensional dynamics depend on the exact entries on the $\boldsymbol{m}^{(r)}$ and $\boldsymbol{I}^{(s)}$ vectors, but not on the individual entries on the $\boldsymbol{n}^{(r)}$ vectors. Instead, $\boldsymbol{n}^{(r)}$ vectors influence the dynamics only through their covariances with the $\boldsymbol{m}^{(r)}$ and $\boldsymbol{I}^{(s)}$ vectors. Thus, all components of $\boldsymbol{n}^{(r)}$ orthogonal to the subspace spanned by $\boldsymbol{m}^{(r)}$ and $\boldsymbol{I}^{(s)}$ are irrelevant for the dynamics, and removing them leads to an effective connectivity matrix $\boldsymbol{J}^{\text{eff}}$ which captures the minimal features required for obtaining particular dynamics.

Thus, for a rank-$R$ RNN, $\boldsymbol{J}^{\text{eff}}$ is computed as:

$$\boldsymbol{J}^{\text{eff}} = \frac{1}{N}\left(\sum_{r=1}^{R}\boldsymbol{m}^{(r)}\boldsymbol{n}_{\|}^{(r)\top}\right), \tag{22}$$

where each $\boldsymbol{n}_{\|}^{(r)}$ is defined as the orthogonal projection of $\boldsymbol{n}^{(r)}$ on the subspace spanned by the $\boldsymbol{m}^{(r)}$ and $\boldsymbol{I}^{(s)}$.

## B Cognitive tasks

Here we describe the input and output structure for the four cognitive tasks used in this study. In all tasks we use the notations from section 2, considering $N_{in}$ input signals $u_s(t)$ and one target output $z^*(t)$, that can be defined only at specific timepoints of a trial. Durations are given in an abstract time mapping, but tasks are implemented in a discretized time with timestep $dt = 20ms$

**Decision-making task (DM)**. The network receives one input signal $u_s(t)$, equal to Gaussian white noise with standard deviation 0.1 (as for subsequent tasks), added to a mean coherence $\overline{u}$ drawn uniformly from $\pm 0.1 \times \{1, 2, 4\}$. The target output $z^*$ is defined only at the final timestep of the trial, and is equal to the sign of the trial coherence. Each trial starts with a 100ms fixation period with no input, followed by an 800ms stimulus epoch, a 300ms delay epoch and a decision timestep.

**Working memory task (WM).** This task is inspired on the traditional parametric working memory and comparison experimental paradigm. The network receives one input signal, equal to:

$$u(t) = f_1 \delta_1(t) + f_2 \delta_2(t) + \xi(t)$$

where $f_1$ is randomly sampled in $[10, .., 34]$, $f_2 - f_1$ is randomly sampled in $\{-24, -16, -8, 8, 16, 24\}$ with the constraint that $10 \leq f_2 \leq 34$ and $\xi(t)$ is white Gaussian noise. The target output $z^*(t)$ is defined only at the final timestep and equal to the exact value $f_2 - f_1$. Each trial starts with a 100ms fixation period with no input, followed by a 100ms stimulus 1 epoch (where $\delta_1(t) = 1$), followed by a 500ms delay epoch, followed by a 100ms stimulus 2 epoch (where $\delta_2(t) = 1$) and a decision timestep.

**Context-dependent decision-making (CDM).** This task aims at modelling the experimental work of (Mante, Sussillo et al., 2013) [29]. The network receives four inputs, two noisy input signals $u_A(t)$ and $u_B(t)$ defined as for the DM task with independently drawn coherences and noise, and two contextual inputs $u_{ctxA}(t)$ and $u_{ctxB}(t)$ defined as a one-hot encoding of the trial context. The target output during the final task epoch is set to the sign of the coherence of the input indicated by the active contextual cue.

For the application of LINT to a full-rank network, the task was defined in five epochs: a 100ms fixation epoch with no inputs, a 350ms epoch with only contextual inputs, an 800ms stimulus epoch, with both noisy stimuli and contextual inputs, a 100 ms delay epoch and a 20ms decision epoch which is the only one where the target output $z^*(t)$ was defined.

For the application of LINT to electrophysiological recordings, four epochs were used: a 350ms epoch where only contextual inputs were active, a 650ms stimulation epoch, an 80ms delay epoch and a 20ms decision epoch (used for computing task accuracy on the fitted networks). The trained networks were constrained to reproduce neural activity only during the last three epochs. The coherences used in this part of the work were sampled from $\pm\{0.047, 0.15, 0.5\}$ for monkey A, $\pm\{0.07, 0.19, 0.54\}$ for monkey F.

**Delay Match-to-Sample (DMS).** This task reproduces a paradigm where two consecutive stimuli each either of type A or B are presented, and the subject distinguishes between matches (both stimuli of the same type) and non-matches (stimuli of different types). In our models, the stimuli are given through two input signals $u_A(t)$ and $u_B(t)$, with Gaussian white noise centered around 0 or 1 while the corresponding stimulus is active. The task comprises five epochs, a fixation epoch of duration 100ms, a first stimulation epoch of 500ms, a delay epoch of variable duration between 500 and 3000ms, a second stimulation epoch of 500ms and a decision epoch of 1000ms. During each stimulation epoch a single type between A and B is sampled and the corresponding input signal is set to a mean of 1. The target output $z^*(t)$ during the decision epoch is equal to 1 for a match, -1 for a non-match.

## C  Training details and hyperparameters.

**Task-optimized lrRNNs.** The full-rank RNNs were defined following a discretized version of equation (1):

$$\boldsymbol{x}_{t+1} = \boldsymbol{x}_t + \frac{\Delta t}{\tau} \left( -\boldsymbol{x}_t + \boldsymbol{J}\phi(\boldsymbol{x}_t) + \sum_{s=1}^{N_{in}} \boldsymbol{I}^{(s)} u_{s,t} + \boldsymbol{\eta}_t \right), \tag{23}$$

with $\boldsymbol{\eta}_t$ a random normal vector with independent entries and standard deviation 0.05 on each entry, and $\Delta t = 20ms$, $\tau = 100ms$. All networks were defined in pytorch [34] and trained using the ADAM optimizer. We trained networks on 800 random trials, for a certain number of epochs being divided in 25 batches of 32 trials. We considered networks with $N = 512$ units for this part of the work.

For the low-rank RNNs (section 3.1), we trained the $\boldsymbol{m}^{(r)}$ and $\boldsymbol{n}^{(r)}$ vectors. For each task, we found the minimal rank by training networks of increasing rank until they performed the task with more than 95% accuracy. For the DM and CDM tasks, the weights were all initialized following a random Gaussian distribution of standard deviation 1, and 4 for the readout weights $w_i$. For the WM task,

the rank-2 networks were initialized from the SVD of the connectivity matrix of a full-rank RNN previously trained on the task. For the DMS task, this initialization trick was also used, as well as the following shaping procedure: rank-2 networks were first trained on trials with a maximal delay of 700ms, then 1000ms, and finally 4000ms. The learning rates used were of 0.01 for all low-rank networks, and on the order of $10^{-4}$ for full-rank networks.

For the full-rank RNNs (section 3.2), we trained the connectivity matrix $\boldsymbol{J}$. The initial weights were sampled from centered Gaussian distributions with a standard deviation of 1 for input and readout weights, and $\rho/\sqrt{N}$ for connectivity coefficients $J_{ij}$. The values of $\rho$ used in the main text were 0.1 for the CDM task, and 0.8 for the DMS task, with results for other values displayed in sup. fig. 1.

**LINT on data from task-optimized lrRNNs**. For the application of LINT to synthetic data generated from the task-optimized lrRNNs, we generated 800 trials for each task, simulated the trajectories displayed by the task-optimized networks on these trials, and trained identical RNNs, all initialized with random weights, to reproduce these trajectories. The networks were then tested on 800 newly sampled trajectories to obtain the reported values of $R^2$. Given trials $k \in \{1, \ldots, K\}$, timesteps $t \in \{1, \ldots, T\}$ and neurons $i \in \{1, \ldots, N\}$, the global $R^2$ is computed as:

$$R^2 = 1 - \frac{\sum_k \sum_t \sum_i \left( x_i^{(k)}(t) - \tilde{x}_i^{(k)}(t) \right)^2}{\sum_k \sum_t \sum_i \left( \tilde{x}_i^{(k)}(t) - \langle \tilde{x}_i^{(k)}(t) \rangle_{k,t,i} \right)^2} \tag{24}$$

following the notations of section 2.

**LINT on data from task-optimized vanilla RNNs**. For this application of LINT, synthetic data was generated from task-optimized full-rank RNNs by simulating their responses to 800 trials for each of the two tasks. Inferred low-rank networks were trained for 500 epochs on those trials, and then tested on 800 newly generated trials. Reported $R^2$ values were computed with equation (24).

**LINT on electrophysiological data**. Low-rank networks trained to reproduce trajectories were trained on a set of 64 random conditions out of 72, and tested on the remaining 8 conditions to compute reported $R^2$. To compute the task accuracy, a linear decoder $\boldsymbol{w}$ was trained on the decision epoch of the task to report the correct choice on all 72 conditions, and the obtained accuracy was reported.

## D   Context-dependent decision-making mechanism

Here we complete the explanations of the main text detailing the mechanisms by which the full-rank network performs the CDM task.

From a dynamical point of view, low-dimensional visualizations have shown that the neural activity first moves along input-driven directions, before being slowly integrated along a choice axis (fig. 3 and sup. fig. 2). The rank-1 inferred network illuminates this behavior in terms of input and recurrent connectivity: the input-driven direction (TDR stimulus A, stimulus B and context axes) correspond to input vectors of the network, directly inserting the stimulus signals into the network. Choice, however, is generated through the recurrent connections, more specifically through a rank-1 feature of these connections: features that have to be integrated are selected by the $\boldsymbol{n}$ vector (which we call the *input-selection vector* [9]), and are projected onto the $\boldsymbol{m}$ vector, which then encodes choice. Hence, overlaps between the signal input vectors $\boldsymbol{I}^A$ and $\boldsymbol{I}^B$ and the input-selection vector drive the integration of input signal into this recurrent loop.

However, the network selects in a context-dependent manner which of the signals is integrated into the loop. We demonstrate in the main-text how this context-dependent selection relies on two separate populations of neurons. More specifically, these two groups of neurons have different weights on the contextual input vectors ($\boldsymbol{I}^{ctxA}$ and $\boldsymbol{I}^{ctxB}$) which drive them towards different gain regimes: for example, the neurons with the strongest weights on the $\boldsymbol{I}^{ctxA}$ vector are driven towards the saturating parts of their non-linear transfer function in context A (fig. 4b). It happens that these neurons exhibit a positive correlation between their entries on $\boldsymbol{I}^B$ and $\boldsymbol{n}$, necessary for integrating the input B signal into the recurrent dynamics. While these neurons see their gain decreased, the effective overlap between vectors $\boldsymbol{I}^A$ and $\boldsymbol{n}$ as described in equation (19) is decreased, and hence input B cannot be integrated anymore. Through this gain-modulation mechanism, the network is able

to ignore the irrelevant signal during context A. The opposite mechanism naturally happens with another population during context B.

This mechanism has a very consequence for ablation experiments: we have shown in fig. 4c how inactivating specifically the green and purple population decrease performance only in one context at every time. A more detailed picture appears when we look at the specific psychometric matrices of the perturbed networks (sup. fig. 2e). For example, when inactivating population B (green), the network actually keeps integrating input A even in context B: it is able to perform only the context A task. The opposite error pattern appears when population A (purple) is inactivated. These error patterns are exactly retrieved in the full-rank networks, showing that the same neurons have an exactly similar role, even though it can be directly retrieved from looking at the full-rank connectivity.

## E    Delay Match-to-Sample full-rank network reverse-engineering

To display another example of how LINT can be exploited to reverse-engineer mechanisms used by a "black-box" full-rank RNN, we apply it to the networks trained on another task. Here we consider the DMS task, where the network receives two consecutive stimuli chosen among two possible classes A and B, separated by a delay period, and has to output during its response period a positive value if the stimuli were of the same class ("match"), a negative value otherwise ("non-match", see full details in appendix B).

When training full-rank RNNs on this task, it appeared that they could be well approximated by lrRNNs of a rank usually equal to two (fig. 2), irrespective of their training hyperparameters (fig. 1). Here, we focus on a full-rank RNN trained with an initial connectivity of standard deviation equal to $0.8/\sqrt{N}$ and $N = 512$ units, which was well approximated by a rank-2 lrRNN ($R^2$ fit qualities for individual neurons shown in sup. fig. 3a). Indeed, this network, although trained without any constraint on its connectivity, exhibited low-dimensional trajectories as can be seen by performing a PCA (sup. fig. 3b). An idea of how neural geometry enables the networks to perform the task can be obtained by observing projections of the trajectories on spaces spanned by the first principal components. Typically, observing projections on the top 3 components (sup. fig. 3c), it becomes apparent that the network relies on four fixed points, one for remembering that the first stimulus is A during the delay period (middle right), one for remembering that the first stimulus is B (at the left), also used to indicate a match, one to indicate a non-match (at the right) and finally one to indicate only a match A-A (at the right). Stimulus inputs seem to drive activity through transient tunnels from one fixed point to another. This picture provides a certain grasp of phenomena happening in the network, but does not illuminate how dynamics enable activity to correctly jump between fixed points, nor how connectivity enables this dynamical picture to emerge.

The rank-2 network inferred by LINT has its activity constrained by design to a four-dimensional subspace of the neural state-space, spanned by the two input vectors of the network $I^A$ and $I^B$ as well as the two output vectors of the recurrent connectivity $m^{(1)}$ and $m^{(2)}$. These four vectors indeed have a significant overlap with the top four principal components (sup. fig. 3d), with the difference that they disentangle input from recurrent contributions to neural activity. The rank-2 connectivity is also characterized by two input-selection vectors of the recurrent connectivity $n^{(1)}$ and $n^{(2)}$, which will be a key driver of the network dynamics, as well as a task readout $w$.

Projecting the neural trajectories on the subspace spanned by the two output recurrent vectors shows indeed that the activity of the rank-2 network reproduces well that of the original network, and that both networks rely on four fixed points to perform the task in the manner outlined above (sup. fig. 3e). In the absence of inputs, the neural dynamics of the rank-2 network stay confined to the two-dimensional $m^{(1)}$-$m^{(2)}$ space, so that the full autonomous dynamics can be visualized as a vector field, illuminating the dynamical inner workings of such networks (sup. fig. 3f). This reveals the existence of the four stable fixed points in the dynamical landscape, as well as an unstable fixed point at the origin and four saddle points. Moreover, while tonic inputs are received, which is the case during each stimulation period, the dynamics shift to a two-dimensional affine subspace of the neural state space, parallel to the $m^{(1)}$-$m^{(2)}$ plane but shifted along the received input vector. The dynamics while a tonic input is received can thus also be visualized as a two-dimensional vector field (sup. fig. 3g), explaining how inputs drive transitions from one fixed point to another. More specifically, only the two stable fixed point on the lower part of the plane are kept during a stimulation by input A, while dynamics have a slight clockwise rotational component. Hence, if input A is

received during a short stimulation period, they are driven towards the lower right part of the plane, where they can stay in a fixed point during the delay period (red and orange trajectories), and if a second input A is received, they will be driven towards the lower left fixed point. Conversely, under input B stimulation, only the two fixed points on the top of the plane are kept, with a very slight counterclockwise rotational component on this plane (at least under the vicinity of the origin, although not very visible in the plotted field). Hence, if stimulus B is received first, the network will be driven to the top left fixed point. If a second stimulus B is received, the network will stay in that state, whereas if a second input A is received it will be driven towards the top right fixed point (non-match).

These full dynamical landscapes provide us with a step-by-step decomposition of the task trials, and can also lead to predictions for trials taht do not appear in the task (for example we could predict the network behavior if three stimulations are received, or with longer or shorter stimulations). Moreover, the rank-2 connectivity can illuminate how this dynamical behavior emerges from the learned synaptic weights [3, 9]. A detailed account would go beyond the scope of this text, but some insights can be extracted easily: first, looking at the distribution of the weights of all neurons on vectors $m^{(1)}$ and $m^{(2)}$ (sup. fig. 3h), it appears that they separate in four clusters, each scattered around a different mean forming a quadrilateral on this plane. This type of population structure has been shown to enable the apparition of polygons of stable fixed points in the neural state space (see notably [3]). The way in which inputs modify dynamics can also be explained by connectivity features: more specifically, inputs A and B seem to modify the gains of different groups of neurons, as was the case for the contextual cues in the CDM task. Importantly, the neurons that are driven to a low-gain regime by input A (defined as the set of neurons $i$ such that $|I_i^A| > 1$) are characterized by a negative correlation between vectors $n^{(1)}$ and $m^{(2)}$, and neurons driven to a high-gain regime by the same input exhibit a different correlation between these two vectors. Due to this differential distribution, while input A is received the effective coupling between the two latent variables $\kappa_1$ and $\kappa_2$ is modified. A converse situation happens with input B, explaining how input A generates this slight clockwise rotational component in the dynamics, and input B generates an opposite rotation. These local rotations, coupled with correlations between inputs and the recurrent vectors lead to the apparition of the plotted dynamical landscapes. These analyses could be verified by examining gains of neurons and performing ablation studies in the full-rank network, as has been done for the CDM task.

## F    Experiments on $K$-back tasks

To probe the capabilities of the LINT method on tasks that by design require higher-dimensional dynamics, we implemented it on the $K$-back task, inspired from classical paradigms of sequential working memory tasks [62]. In this task, for a certain fixed $K$, networks received a random number $P \geq K$ of stimulations in $\{-1, +1\}$ and had to output the $K$-th last stimulation received. In particular, for $K = 1$, this task corresponds to the classic running-memory flip-flop task [55]. Stimulations were received through a single input signal $u(t)$ transmitted by an input vector $I$ and outputs were obtained through a linear readout following eqs. (1) and (6). Each stimulation lasted for 50 ms, and was separated from the next one, or the response period, by a 50 ms delay. We implemented the task following the methods outlined in appendix C, with 512-neuron networks. $P$ was capped at 12.

This task requires the networks to implement an internal memory with a capacity of at least $K$ bits, it is thus reasonable to believe that a rank at least $K$ is necessary to perform it. We first directly sought to validate these intuitions by training networks of increasing ranks to behaviorally perform the task, for $K$ between 2 and 6. We observed that indeed the rank increased as $K$ did, and that a rank equal to $K$ seemed to be sufficient to implement the $K$-back task (sup. fig. 9a).

We next probed whether LINT could be applied to full-rank networks trained on the $K$-back task, despite the higher dimensionality. We thus trained full-rank networks on the task, for $K$ from 2 to 6 (and an initial connectivity of standard deviation $\rho/\sqrt{N}$ for $\rho = 0.8$), and fitted their trajectories with low-rank networks of increasing ranks. Again, we found that full-rank networks trained on the $K$-back task appeared to be imitated well by low-rank networks with a rank as low as possible, i.e. $K$, both in terms of trajectories and behavior (sup. fig. 9).

# G   Supplementary figures

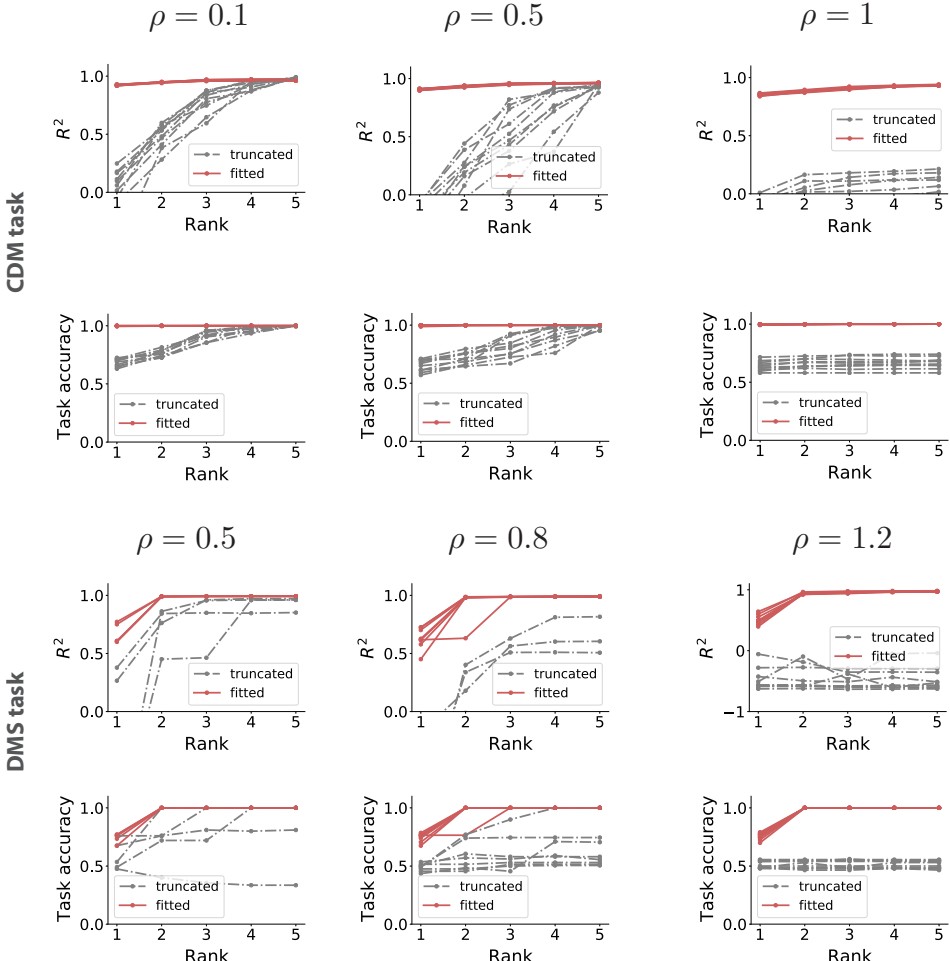

Supplementary Figure 1: For different values of $\rho$, where the standard deviation of initial recurrent weights is $\rho/\sqrt{N}$, we train 10 unconstrained networks, and either fit low-rank networks to their trajectories with increasing ranks or truncate their connectivity matrices to the same rank. The obtained $R^2$ similarity between original and fitted trajectories and task accuracy when plugging low-rank networks to the original readout are illustrated with a different line for each original unconstrained network. Top 2 rows: experiments on Context-dependent decision making task. Bottow 2 rows: experiments on the Delay Match-to-Sample task. Note that with higher initial random recurrent weights, unconstrained networks tend to go to what has been termed as the "lazy" training regime, with potentially higher dimensional trajectories with respect to the the "rich" training regimes for smaller initial weights [15]. This is visible through the poor performance of truncated connectivity matrices, but does not harm the effectiveness of our method.

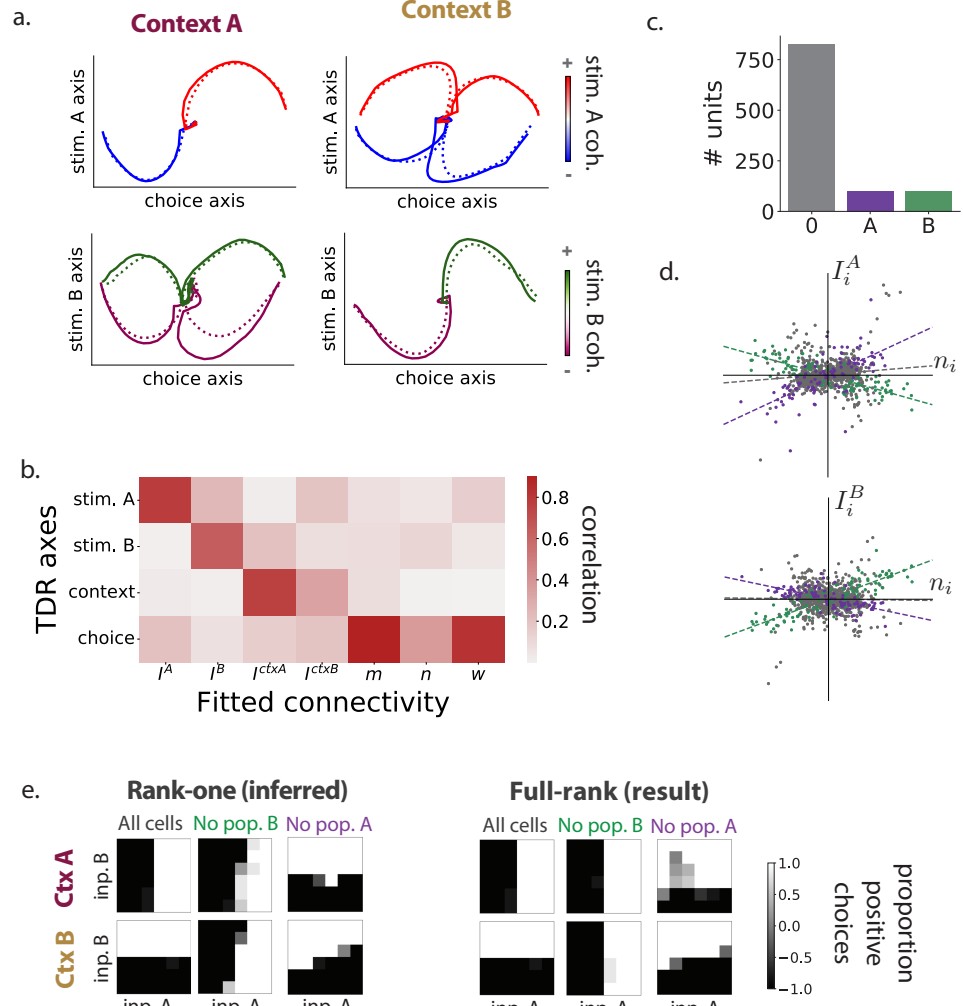

Supplementary Figure 2: Additional figures on LINT applied to a full-rank network performing the CDM task (exhibited in figs. 3 and 4). **a**. Low-dimensional projections of trial-averaged population trajectories for several combinations of context, stimuli A and B and choice, as in fig. 3 in the original full-rank network (full lines) and the inferred rank-1 network (dashed lines), projected on axes found by targeted dimensionality reduction (TDR) [29] applied to the full-rank network. **b**. Correlation between the four axes found by TDR on the full-rank network and the connectivity axes inferred by LINT. **c**. Number of units assigned to each of the three populations used for the reverse-engineering. Here, we manually defined population A as the 100 units with the strongest absolute context A input weight in the rank-1 network (see fig. 3a top), and population B as the 100 units with the strongest context B input weight not in population A. Applying Bayesian GMM clustering with 3 components and a strong mean precision prior gives very similar results. **d**. For the inferred rank-1 network, joint distributions of connectivity weights on the input vector $\boldsymbol{I}^A$ and $\boldsymbol{n}$, as well as on $\boldsymbol{I}^B$ and $\boldsymbol{n}$. For each population, linear regression lines are plotted. **e**. Psychometric response matrices in each context for all combination of stimulus coherences, for the inferred and original network when they are left unperturbed or after lesioning populations A and B. Unperturbed matrices indicate expected behavior. One can observe that inactivation of population B leads the networks to always behave as if in context A (losing its capacity to perform in context B), whereas the opposite phenomenon happens when population A is inactivated.

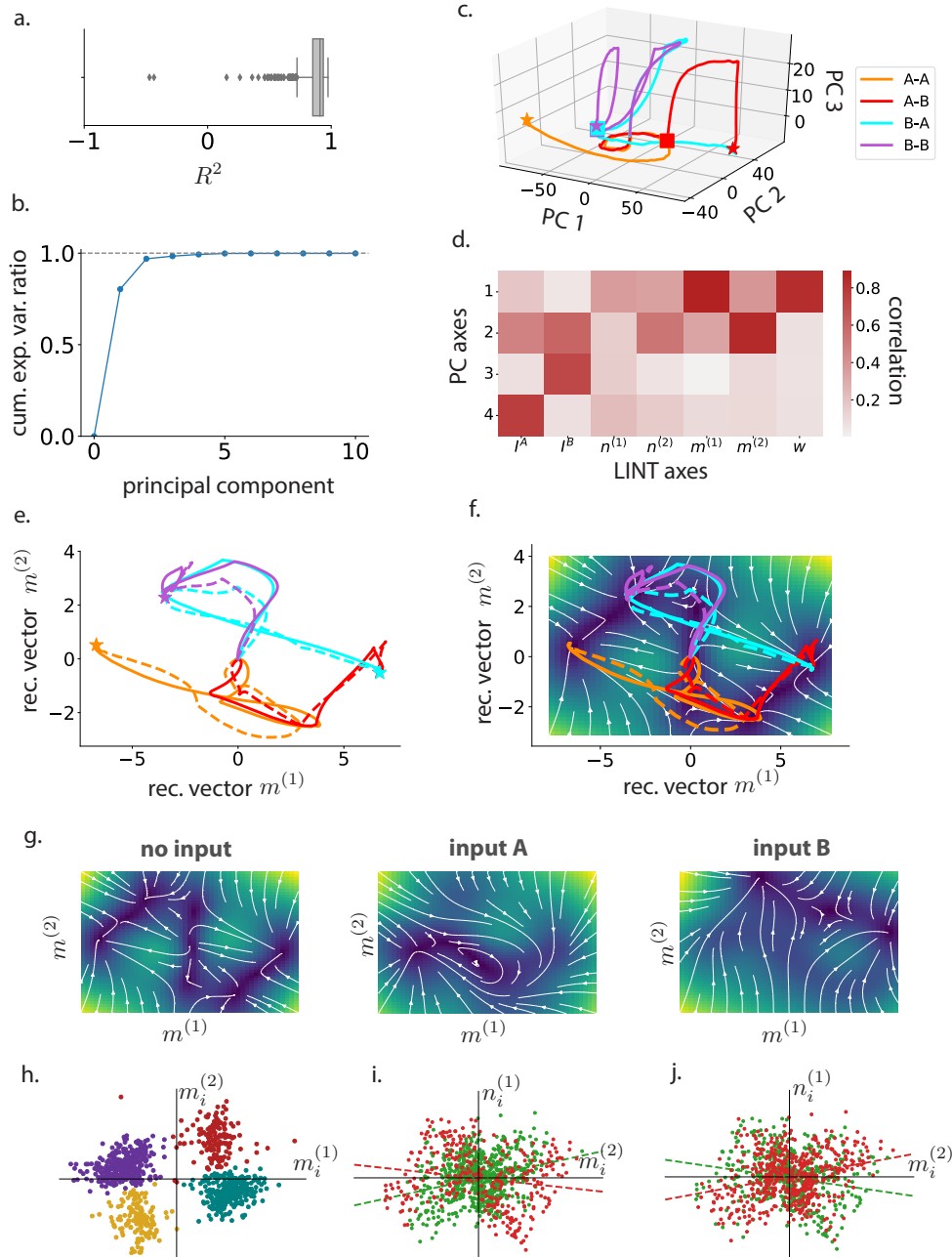

Supplementary Figure 3: LINT applied to a full-rank RNN trained on the DMS task. A rank-2 network was inferred and is analyzed in this figure (see Appendix E for details). **a**. Boxplot representing the distribution of $R^2$ fitting values for individual neurons. **b**. Cumulative explained variance ratio for top 10 principal components of a PCA applied to trajectories of the full-rank network. **c**. Trajectories in the four possible task conditions in the original full-rank network, projected on the top 3 principal components (squares: delay period, stars: end of trial). **d**. Correlation between axes inferred by a PCA on the full-rank trajectories and connectivity axes of the inferred rank-2 network. **e**. Trajectories of the full-rank network (full lines) and the rank-2 model (dashed lines) in the four task conditions, projected on the two recurrent connectivity output vectors $\boldsymbol{m}^{(1)}$ and $\boldsymbol{m}^{(2)}$ (same colors as in c). **f**. Same trajectories, superposed on the vector field representing autonomous dynamics in the rank-2 RNN. Colors indicate speed of the dynamics (blue: slow, yellow:fast). **g**. Vector fields representing the dynamics on the $\boldsymbol{m}^{(1)}$-$\boldsymbol{m}^{(2)}$ plane. **h-j**. Connectivity parameter distributions on the rank-2 models... **h**. on the two recurrent output vectors $\boldsymbol{m}^{(1)}$ and $\boldsymbol{m}^{(2)}$ - four populations can be identified by GMM clustering. **i**. on the connectivity vectors $\boldsymbol{m}^{(2)}$ and $\boldsymbol{n}^{(1)}$ - low-gain neurons while input A is received in red, others in green with overlaid linear regressions for both groups. **j**. on the same vectors, with low-gain neurons while input B is received in red, others in green.

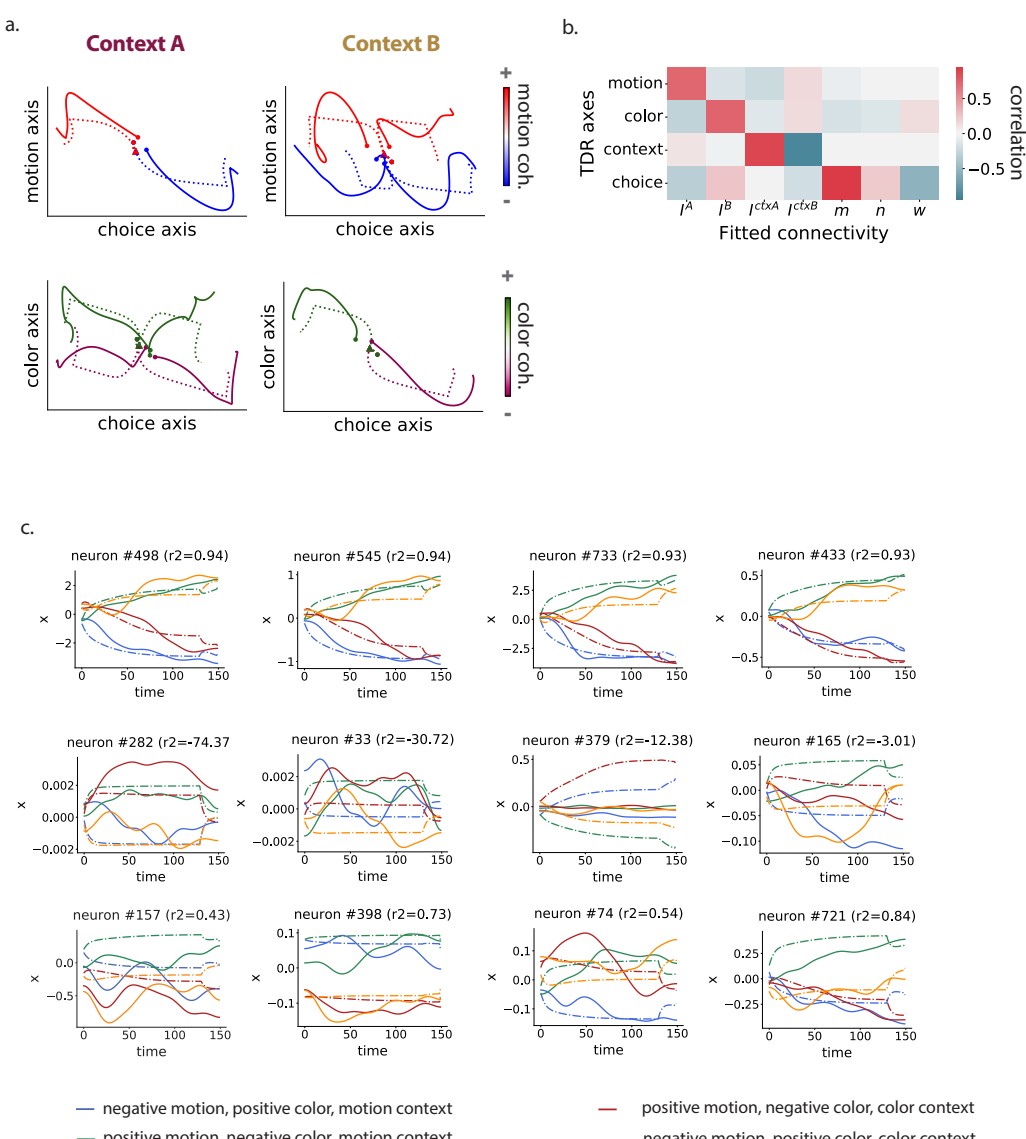

Supplementary Figure 4: Additional illustrations of LINT applied to electrophysiological recordings in monkey A. **a**. Two-dimensional projections of trial-averaged population trajectories for several combinations of context, choice, and motion or color coherence (indicated by the color code), as in fig. 5 c, in the recorded data (full lines) and the rank-1 model (dashed lines), projected on axes inferred by TDR. **b**. Correlation coefficients between axes identified by TDR and LINT connectivity axes. **c**. Pre-processed data responses and rank-1 model responses for individual neurons to 4 different task conditions (uniquely identified by a context, a color coherence, and a motion coherence. Strongest coherences displayed here). Top row: four best fitted neurons. Middle row: four worse fitted neurons. Bottom row: four randomly selected neurons.

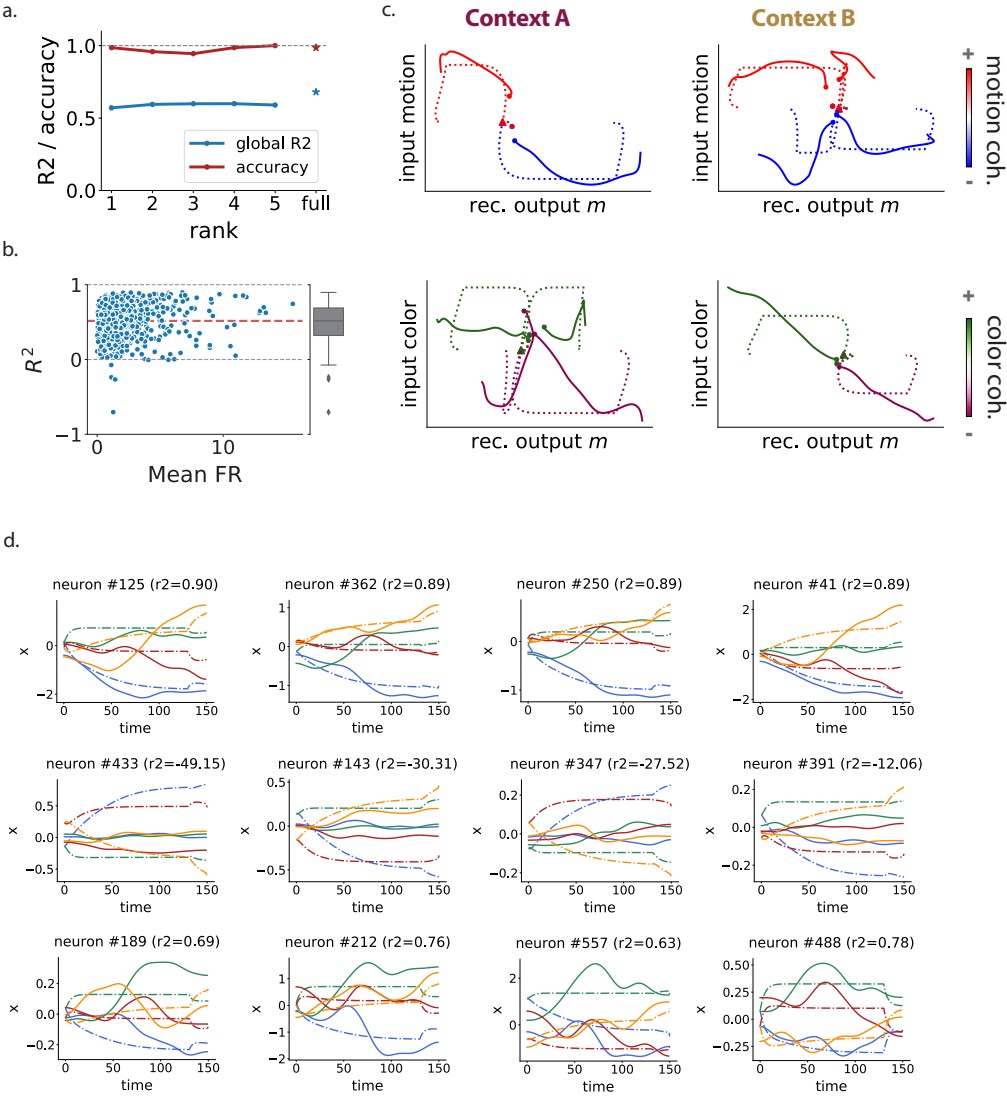

Supplementary Figure 5: LINT applied to electrophysiological recordings of a second monkey (monkey F) performing the same task. **a-c**. Same as fig. 5 for monkey F data. For panel b, 13 neurons for which $R^2 < -1$ do not appear, all having a mean firing rate of less than 3.8 Hz. **d**. Same as sup. fig. 4 for monkey F, with the 4 best fitted neurons, the 4 worse fitted neurons, and 4 randomly selected neurons.

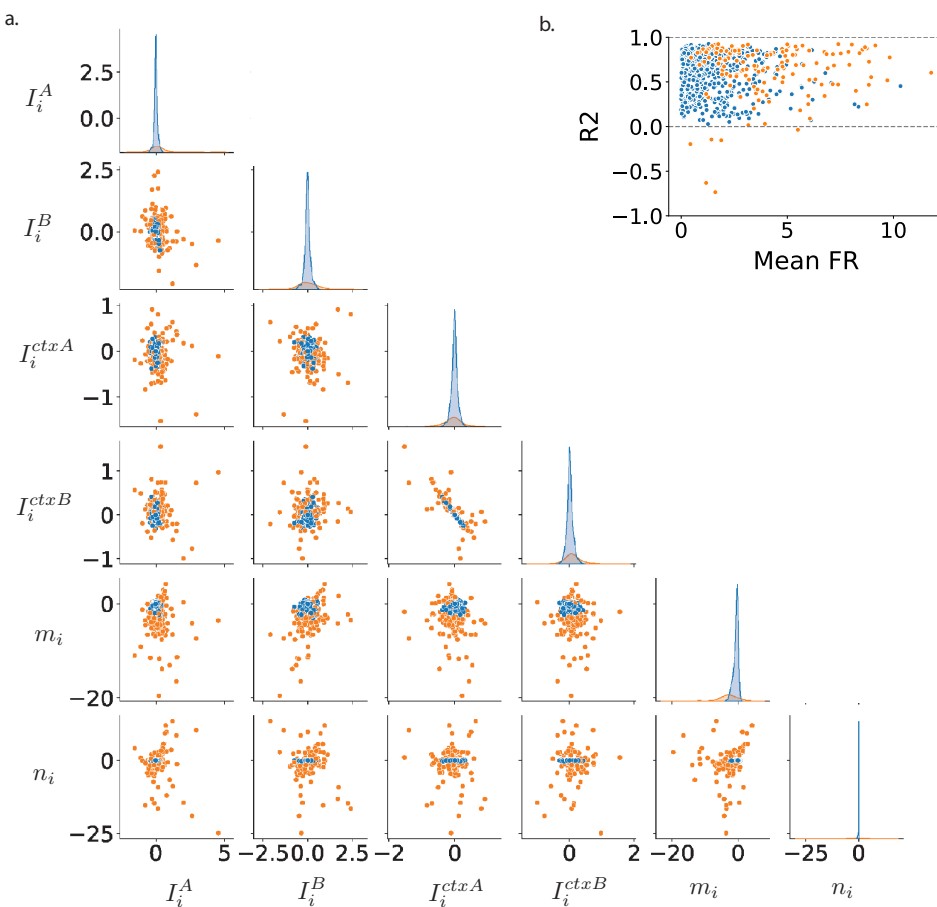

Supplementary Figure 6: Distribution of learned connectivity parameters for monkey A. **a**. Full six-dimensional distribution of the inferred weights on the input and recurrent connectivity vectors of the rank-1 model, plotted through two-dimensional and one-dimensional marginals. GMM clustering can identify two groups of neurons: large-weight neurons (orange) and small-weight neurons (blue). Inactivating the blue population does not affect task performance of the network. **b**. Same clusters visualized in the mean firing-rate - $R^2$ point cloud (fig. 5a).

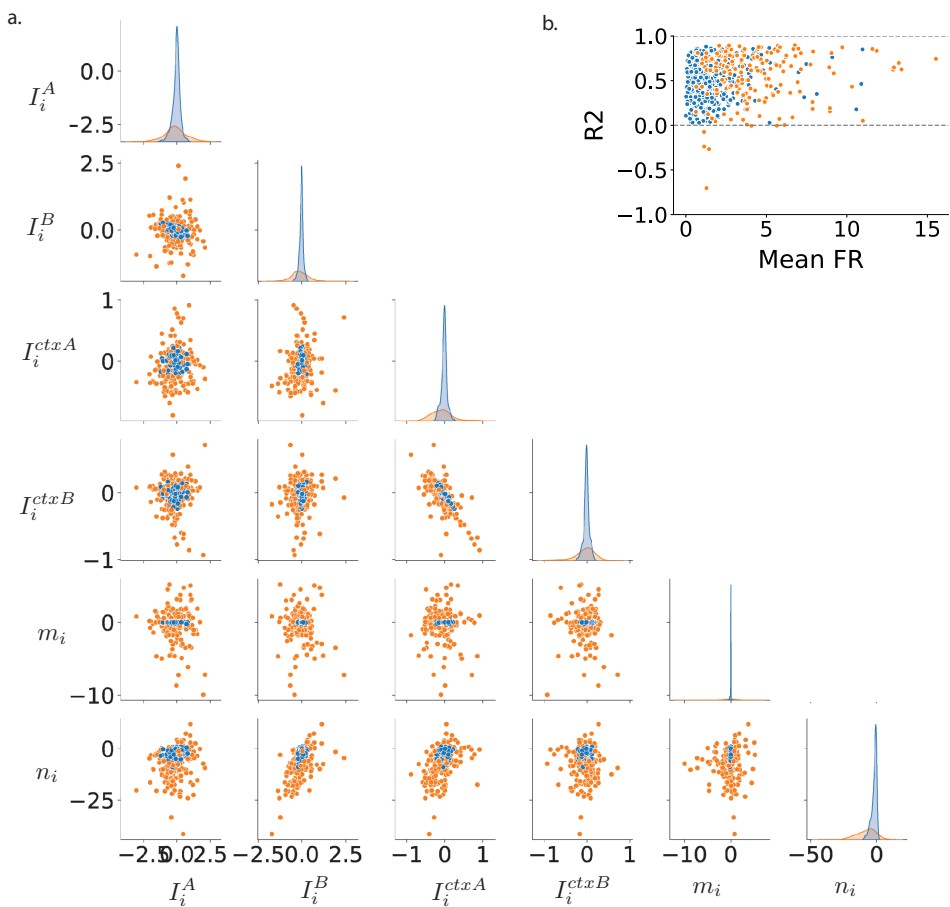

Supplementary Figure 7: Same as sup. fig. 6

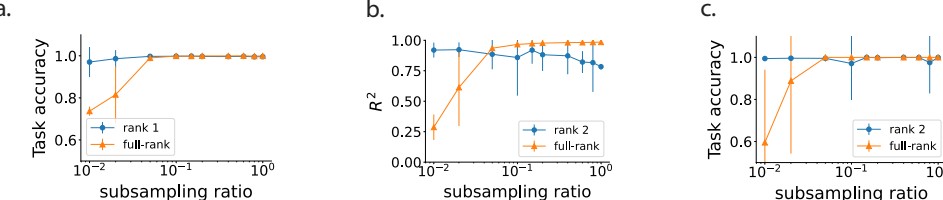

Supplementary Figure 8: Additional results on subsampling experiments. **a**. Task accuracy for the same fitted networks visualized in fig. 2f. **b-c**. Similar experiment as that described in fig. 2f, this time where trajectories were generated from a full-rank network trained to perform the DMS task, and respectively full-rank and rank-2 networks were fitted to random subsamples of neurons of the original network. Error bars: mean $\pm$ std over 10 random subsamples for each ratio value.

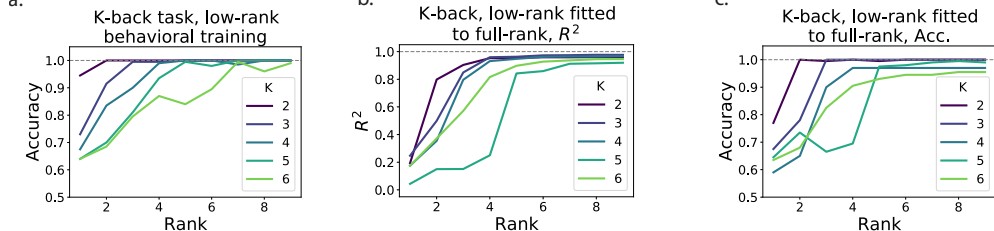

Supplementary Figure 9: Networks trained on the $K$-back task (see appendix F). **a**. Final accuracy of low-rank networks of increasing rank trained to perform the task, for $K$ between 2 and 6. Notice that a rank $K$ seems necessary and sufficient for the $K$-back task. **b**. $R^2$ values for the trajectories of full-rank networks trained on the $K$-back task, and networks of increasing rank trained to reproduce their trajectories, for $K$ between 2 and 6. **c**. Final task accuracy for the same low-rank networks.