# OpenReview forum: "Extracting computational mechanisms from neural data using low-rank RNNs"
_NeurIPS.cc/2022/Conference — NeurIPS 2022 Accept_

### Official Review · Reviewer_ittd · 2022-07-08

**Rating:** 6
**Confidence:** 4
**Soundness:** 3 good
**Presentation:** 4 excellent
**Contribution:** 3 good

**Summary:**

This paper introduces low-rank RNNs, parameterized by the left and right eigenvectors of the connectivity matrix. The authors first show on simulated full-rank vanilla RNNs trained to perform some cognitive tasks that low-rank RNNs can accurately extract the essential dynamical and task structure learned by the full-rank systems. They thereby offer a computationally interpretable low dimensional representation of task activity. The authors then test their approach on neural recordings from two monkeys performing a context-dependent decision making task. Again the trained low-rank RNNs seem to extract the behaviorally relevant low-dimensional dynamics, at the same time producing synaptic weight distributions reminiscent of those observed in the brain.

**Questions:**

1) One may perhaps still ask what is really novel about this approach, given that low-rank RNN-type models have been employed in computational neuroscience for a number of years now to obtain insight into neural dynamics and task performance. I assume the authors are the first to actually train such a model directly on simulated and real neural data and to systematically explore its abilities in identifying the dynamical structure?

2) Directly inferring a lrRNN seems to be more powerful than training a full-rank RNN and using SVD posthoc. But why actually? Can the authors offer some theoretical insight why this is more efficient? Does this simply have to do with the much lower number of parameters that need to be inferred on lrRNNs, easing the training process?

3) The authors call the CDM and DMS tasks ‘complex’, but compared to many real world tasks and those in use elsewhere in the ML community I find them almost simplistic. Could the low-rank structure be an ‘artifact’ or result of the very simple nature of these tasks? For instance, say you were to construct a very high-dimensional space where dimensions are somewhat modulated by task features like inputs, would a lrRNN always infer this same structure cos it’s really a property of the task, not of the recorded neural activity?

I think this is a relevant question since I was surprised that, despite the very simple nature of the tasks, hundreds of neurons were used for the full-rank RNNs trained on the tasks. More importantly, the neural recording data sets appear to consist likewise of hundreds of neurons which were however drawn *independently* from different sessions and trials? This appears to imply that the neurons are *not* truly dynamically connected, and do not have any correlations beyond that induced by the task structure. Thus, the inferred lrRNNs may simply reflect this task structure but not really the processing and (missing) dynamics that goes on in the neural population during a task.

4) A related question: As I assume hundreds of RNN units are not really necessary for learning these tasks, couldn’t we just train a much lower-dimensional system to begin with and thereby obtain a much more parsimonious representation of dynamics? I would guess for these tasks 5 units may actually already do the job?

5) Is the phenomenon of mixed selectivity in cortical networks at odds with the low-rank structures identified?

6) I would like to see more examples of original and fitted neural firing rates and neural trajectories. For instance, in sect. 3.1 the claim is made that essential connectivity features and neural trajectories were well reproduced by the lrRNN, but Table 1 shows only a single summary statistic w/o confidence limits. Same goes for the real neural recordings, more unit examples that illustrate neural behavior on different trials of the task would be great.

7) Since Fig. 3-5 appear to show just examples of a single training run, I would like to see more stats on how robust the training process is, i.e. how likely the same structures are to be identified again when starting from many different initial conditions for parameters?

I think point #3 is really my main issue. In any case it would have been nice to see the application of this method to at least one other data set with a very different task. But given the third point, in my mind it’s essential to see an application on a higher-dimensional set of truly simultaneously recorded neurons.

Minor points:

- Line 129: Universal approximation properties in my mind don’t apply here: To assure this, you would need to allow for a an arbitrarily long basis expansion, commonly implying that J would need to be of full rank and, in addition, you may need many more latent dim. than observed variables.

- My understanding is that for any real physiological data set one would need to re-train the lrRNN for a number of different R’s?

- Line 146-147: says that at least rank of 9 is necessary, but Fig. 2 only shows up to rank 5.

- Would be good to have connectivity matrices from the Suppl. in main text. I think they add a lot to the understanding of the inner workings of the trained lrRNNs.

- I’m somewhat puzzled why so many units are used for lrRNNs to begin with, and then kept after training? The results on the CDM, for instance, seem to imply one just needs 5 units?

- Does the lrRNN trained on the monkey data have as many dimensions as units? Isn’t this a constraint that could also prevent detection of latent structure in the data, since there are no latent variables in the lrRNN that could account for unobserved processes?


**Limitations:**

Partly, see questions above.

**Strengths And Weaknesses:**

Strengths:
Generally speaking, I think this is a powerful approach that may have important implications in neuroscience. Building a low-rank structure right away into the models used for inference may profoundly ease detection of computational mechanisms from data.

Weaknesses:
See detailed questions below. My major point probably is that I don't think it's clear whether the model really extracts the neural dynamics and how an animal is truly solving the task, or whether it merely finds a parsimonious solution to the task.

---

> ### Author Response · Authors · 2022-08-01
> **Answer to Reviewer itdd**
>
> 1. The reviewer is absolutely correct, we believe this is the first study to train low-rank RNNs to fit neural activity. See “Novelty” in the General Rebuttal.
> 2. The objective of an SVD is to optimize the low-rank reconstruction error on the connectivity matrix J. This objective does not take into account the input and output vectors. We believe it is therefore simply not relevant to approximation of the network dynamics. Note that other works have tried this, for example (Krause et al. biorXiv 2022) or (Schuessler et al. NeurIPS 2019). We will clarify this point in the Discussion of the revised paper.
> 3. We agree that the low-dimensionality of the tasks is a key concern, and we have focused on addressing it by studying an additional task. See  "low-dimensionality of tasks" and "limitations" in the General Rebuttal. These new results will be included in the revised manuscript. Concerning the non-simultaneous character of the recordings used here, we simply tackle it by considering only trial-averaged data, which is a classical approach with this kind of data. Although we have not yet had the occasion to perform analyses on datasets of simultaneously recorded neurons, this is a direction that we wish to explore in future work, and in which single trials could potentially be fitted.
> 4. This point is particularly relevant to this line of work. Indeed the reviewer is correct that a 5-unit network is sufficient to perform the CDM task, although it is not clear how such a network will map to brain-like computations, distributed over large networks. Our approach specifically provides a method for identifying a small network of latent variables (with potentially multiplicative interactions) that emerges as an effective description of a large vanilla RNN. The rather large number of units in the vanilla RNN allows us to perform a principled mean-field analysis.
> 5. Low-rank RNNs in fact precisely exhibit mixed-selectivity (see Mastrogiuseppe & Ostojic, Neuron 2018). The original goal of this class of models was to precisely reconcile the observations of mixed-selectivity with observations of low-dimensional dynamics.
> 6. This is a relevant suggestion, and we will include more single unit examples in the revised manuscript.
> 7. Reproducibility of the low-rank structures is addressed through the simulations of Fig. 2 and SI Fig. 1. It is not guaranteed that the same population mechanism would consistently be retrieved across different seeds and/or hyperparameters in the original full-rank network. Our goal  to present the reverse-engineering analysis as a proof-of-concept of how our method can help probe deep mechanistic underpinnings of trained black-box networks (see also the DMS example in Sups). Note that the full-rank network analyzed in these sections was selected randomly, i.e. not cherry-picked.
>
> Minor points:
> - Indeed, universal approximation would involve the presence of additional hidden units. We reworked the sentence to avoid confusions.
> - Indeed, the implication is that for any dataset the optimal rank has to be searched for by trying different values.
> - This was a typo, now corrected. Thanks for spotting it.
> - We are not sure to understand which matrices are referred to here, but would be glad to include them in the main text if space permits.
> - See points 4 and 5 above.
> - We made the choice to match units of the model to recorded units on a 1-to-1 basis, without adding more hidden units mainly because we aim at understanding the computations that arise at the level of the recorded circuit. By providing hidden units to the model, we were concerned that it could "hijack" the computations by delegating them to the hidden units, failing to teach us what dynamics can arise in the area being studied.

---

> > ### Comment · Reviewer_ittd · 2022-08-08
> > **Thanks for rebuttal**
> >
> > I thank the authors for their thoughtful responses and revision. I’m still in favor of this work, although I think my main point (3) has not really been addressed. That is, the K-back task is perhaps helpful in further confirming that the authors’ method works well. But it doesn’t give me more reason to believe the model really retrieves neural dynamics underlying task performance rather than simply the task structure itself. Perhaps this is too difficult to answer w/o further experiments on simultaneous recordings, i.e. w/o averaging across independently drawn cells.
> > I still think it’s a valuable contribution. But this prevents me a bit from raising my score further.

---

### Official Review · Reviewer_db5J · 2022-07-11

**Rating:** 5
**Confidence:** 3
**Soundness:** 3 good
**Presentation:** 2 fair
**Contribution:** 2 fair

**Summary:**

In this paper, the authors use low-rank RNNs for modelling neural data. The connectivity matrix is approximated using a low-rank representation, which is then inferred by minimising the loss between the predicted neural outputs and the target ones. In task-optimised RNN, relevant parameters to are trained based on task targets. The method is then applied to various cognitive tasks to validate and extract insights.


**Questions:**

- The assumption is 'm' vectors are orthogonal to each other (and similarly for 'n'). How is this constraint enforced during training?

- 3.2 assumes that m,n, I, and w are all zero-centred. Please expand on why this is a valid assumption.

- One can first obtain a full-ranked connectivity matrix and then approximate it using the low-ranked method suggested here (similar to what is done for SVD in the paper) rather than learning a low-ranked approximation directly. How is this alternative compared to what is suggested in the paper? On a similar note, one can train a truncated SVD directly (rather than making the approximation post-training); how is this method's performance compared with the proposed method?

**Strengths And Weaknesses:**

Strengths:
The models used are rather simple, which benefits from good interpretability.
The experiments considered are interesting and elucidate different aspects of the model.

Weaknesses:
Novelty. The novelty of the paper in terms of the model is very limited. The paper shows that low-rank RNNs can help analyse neural data, but the contributions of the work beyond these experimental evaluations and applications of the low-rank RNNs are unclear.

Minor weaknesses:
Presentation: From the main text, it is unclear how \kappa and \v enter the analysis.
Similarly, the role of section A.2 is rather unclear. Is it to justify the effective connectivity metric?

---

> ### Author Response · Authors · 2022-08-01
> **Answer to Reviewer db5J**
>
> We thank the reviewer for their time and their positive comments about our work. We address their concerns in the point "novelty" of our general rebuttal, reproduced here for convenience:
> This work builds on previous studies that fall into two classes: (i) theoretical analysis of low-rank RNNs trained or designed to perform specific tasks (Mastrogiuseppe & Ostojic 2018, Dubreuil et al 2022, Schuessler et al 2019 notably); (ii) data analyses of full-rank RNNs trained to fit recorded neural activity (Rajan et al. Neuron 2016, Perich et al biorXiv 2021). The key novelty of this work is that we combine these two approaches: we train low-rank RNNs to fit neural activity, and then use the low-rank theory to analyze and interpret the obtained networks. Methods for automatic discovery of interpretable models of neural activity are an important goal for neuroscience and ML research, and we believe our work provides a significant contribution to this literature.
>
> We hope this response (and the assessment of other reviewers) alleviate the reviewer's concerns about the contributions of our work to the field. We would therefore like to humbly ask if they might consider raising their score above the acceptance threshold.
>
> Minor weaknesses: the purpose of introducing the dynamical framework, with variables $\kappa_r$ and $v_s$ is to demonstrate the necessary low-dimensionality of those frameworks, and how they align with the general line of work modeling neural computations through low-dimensional dynamical systems (Vyas et al. 2020). In general, $\kappa_r$ and $v_s$ parametrize the low-dimensional trajectories illustrated, for example in Fig. 3 and Fig. 5, which we will clarify in the corresponding captions. Appendix A.2 mostly justifies mathematically the concept of effective connectivity, which shows how multiple RNNs with different connectivities can lead to the same trajectories, and justifies our use of low-rank models to simplify this degeneracy.
>
> Questions:
> - How is the orthogonality of ‘m’ vectors enforced? During back-propagation, the weights $m_i$ and $n_i$ are trained, without constraining the $m^{(r)}$ vectors to be mutually orthogonal (same for the $n^{(r)}$ vectors). At the end of training, we compute the connectivity matrix J from the obtained m and n vectors. We then perform an SVD on J, which leads to the final $m^{(r)}$ and $n^{(r)}$ vectors, where the m vectors are mutually orthogonal (and same for the n vectors).
> - Zero-centered parameters: the general method does not assume centered weights for all vectors (and for the DMS task in sup. material the parameters are not zero-centered). The interpretation of the CDM task in section 3.2 is partly based on the empirical observation of centered weights (see fig. 4a). The main results remain valid if the weights happened to not be centered during training (in particular all section 3.2.1, identification of populations, and gain modulation mechanisms).
> - we are not entirely sure to have understood the question, but believe the reviewer suggests  a 3 step protocol that would involve: neural data -> full-rank model -> low-rank approximation of the full-rank model. This protocol would add a step and could lead to more information loss between the original data, and the final low-rank model. Nevertheless, we do test our method with full-rank instead of low-rank models, in fig. 2f, and in fig. 5a notably.

---

> > ### Comment · Reviewer_db5J · 2022-08-07
> > **reply**
> >
> > Thank you for your response. I still find the contributions not high, but I agree directly modelling neural activities is an interesting direction. In light of this, I increased my score.

---

### Official Review · Reviewer_8DBW · 2022-07-11

**Rating:** 7
**Confidence:** 4
**Soundness:** 3 good
**Presentation:** 2 fair
**Contribution:** 3 good

**Summary:**

This paper describes results from a method for estimating low rank RNNs from observed trajectories.  The method itself is just gradient descent on the set of weights that specify a low-rank RNN to match observed trajectories.  The paper demonstrates that the method can recover the effective connectivity of known low-rank RNNs, does a good job describing the behavior of general RNNs trained on a set of four tasks, and does a reasonable job recovering neural trajectories from NHPs performing one of those tasks.

**Questions:**

It would be very informative to include a situation where the dynamics are in fact relatively high-dimensional.  Note that a full-rank RNN trained on the tasks used here does not necessarily exhibit the growth in dimensionality.  One possibility is to compare to an RNN trained to distinguish a large number of temporal intervals.

**Limitations:**


The paper should do a better job at pointing to the computational limitations of low-rank RNNs and their possible limitations in describing brain data and cognition.



**Strengths And Weaknesses:**

Strengths

Given the widespread interest in RNNs in machine learning and neuroscience, this method may find broad applications in both of those fields.  The low-rank description is more readily interpretable.

Weaknesses

I think the paper seems to take as a given that low rank RNNs can provide a veridical description of brain dynamics and learn many tasks.  This is not at all well established.  The tasks chosen here are well-described by low-dimensional RNNs, but many tasks and natural networks appear not to have that property.  For instance, Cueva et al., (2020, PNAS) estimate the dimensionality of neural populations during the delay period of a variety of experiments (see especially their figure 7).  They find a number on the order of 5-10 for the dimensionality over a few seconds.  Leaving aside technical questions about whether a particular number is meaningful, one can read their result as decelerating but growing without bound (e.g., dimensionality $\propto \log t$), or as saturating at some time $T$.  The first interpretation argues that a low-rank description is not appropriate---or at least that rank grows to be much larger than 10.  The other possibility---that dimensionality saturates at some time T---suggests it would be impossible for the network to distinguish times $> T$.  But that seems inconsistent with the data (e.g., Lewis & Miall, 2009, Trans Royal Soc B).

Kind of a minor issue, but submitting a set of images instead of a searchable PDF made the job of reviewing this paper more difficult than it needed to be.

---

> ### Author Response · Authors · 2022-08-01
> **Answer to Reviewer 8DBW**
>
> We are very grateful to the reviewer for their very positive comments and constructive suggestions.
> To address the comment and question concerning the dimensionality of the task, we have followed the reviewers’ suggestion and examined the K-back task. See General Rebuttal for details (point “Low-dimensionality of tasks”).
>
> We hope this addresses the reviewer’s concern. We would be glad to add those results to the revised paper.

---

> ### Comment · Reviewer_8DBW · 2022-08-07
> **Response to rebuttal**
>
> I have read the rebuttal and remain positive.  Adding the K-back task with a wide range of K's would be a valuable contribution.  The authors may want to consider (and perhaps comment) on the observation that the Cueva paper sees increasing dimensions in experiments with a fixed delay interval.

---

### Official Review · Reviewer_98wL · 2022-07-12

**Rating:** 6
**Confidence:** 4
**Soundness:** 3 good
**Presentation:** 4 excellent
**Contribution:** 3 good

**Summary:**

This paper studies the problem of modeling dynamical systems using low-rank RNNs. The paper motivates a number of domains in which low-rank RNNs have given insight into the computations underlying a particular task. The paper proposes directly fitting low-rank RNNs (meaning the connectivity matrix in the RNN is restricted to be low-rank). The paper demonstrates how this restriction allows one to learn the correct effective connectivity for a number of synthetic tasks, and also applies the method to recordings from macaque prefontal cortex while the animals performed a context-dependent decision making task. This method reveals that a rank one network is sufficient to capture the neural activity.

**Questions:**

- Why use an explicit (hard) rank constraint, rather than an implicit (soft) constraint, for example by penalizing the nuclear norm of the connectivity matrix? My understanding is that the latter is admits an easier optimization problem, as the nuclear norm is the convex relaxation of a hard rank constraint.
- There are multiple gray and red curves in Fig 2. What do they correspond to? Different random seeds?
- Why is there no equivalent of Fig 2f. for the DMS task?

**Limitations:**

Yes

**Strengths And Weaknesses:**

Strengths:
1. The paper is well written, and does a good job of motivating the problem and summarizing relevant work.
2. The paper's focus on low-rank structure as an important inductive bias for neuroscience is not particularly new, but is nonetheless important. For example, Fig 2f. is a powerful reminder of the utility of the appropriate inductive biases for systems neuroscience, where we are (generally) only able to sample a tiny fraction of the relevant neurons participating in a given task.
3. The experiments on synthetic tasks as well as the application to neural data are clear and high quality.

Weaknesses:
1. The main contribution of the paper is essentially advocating for a type of regularization (low-rank structure), one that has been discussed plenty in the literature (for example by Mastroguiseppe and Ostojic, which is already cited by the paper). While important, the overall significance and novelty does not seem that high.
2. It is not clear how one generalizes the method to RNNs beyond the ones described by equation (1). For example, what about RNNs with multiple weight matrices (such as gated architectures: GRUs or LSTMs)? These are commonly used in machine learning and neuroscience, I am curious if or how the authors would apply their method to those networks.
3. The paper studies tasks which can readily be solved by a low-rank vanilla RNN. But the space of tasks that we are interested in understanding is much greater! What happens if you apply this method to a dynamical system that is solving a more complicated task? For example, one that does not admit a low-rank solution? It would be good to get a sense of what to expect from the method in these scenarios.

Minor suggestions for improvement:
- Should explain the "effective connectivity" in the main text. It is used throughout, and thus seems important enough to put in the main text (currently the explanation is in appendix A.4).
- The entire PDF is low resolution (for exmample text) which makes it hard to read.
- This is a nitpick, but replace $J^{eff}$ with $J^{\text{eff}}$. That will prevent the "eff" from being italicized (which should only be done for mathematical variables).

Overall, I am not so worried about point (1) under weaknesses, but look forward to see if the authors can address (2) and (3) in the rebuttal.

---

> ### Author Response · Authors · 2022-08-01
> **Answer to Reviewer 98wL**
>
> We thank the reviewer for their time and effort in reviewing our manuscript, and their positive and relevant remarks. Their concerns are overall aligned with those of other reviewers, and we refer to our general rebuttal where they are addressed, and answer to some specific points here.
>
> Responses to weaknesses:
> 1. See point "novelty" in the General Rebuttal.
> 2. See point "other architectures" in the General Rebuttal.
> 3. What happens for more complex tasks? This is a very interesting question, to which we devote new analyses in our current revision, on a task that is higher dimensional by design, the K-back task.
> See "low-dimensionality of tasks" in the General Rebuttal for details.
>
> Minor suggestions: these are very relevant and we have now corrected most of them. We do not understand what technical issue has caused the registration of the pdf as images, and are very sorry about it. We had to move the explanation of the effective connectivity correlation by lack of place, as they are a specific point. This measure does not appear except in section 3.1.
>
> Questions:
> Why use a hard constraint on rank rather than nuclear norm?
> We found empirically that it is easier to control the rank by enforcing it with a hard constraint, than by a soft constraint. More systematic comparisons with nuclear norm regularizations however remain to be performed.
> What are the multiple lines in Fig 2?
> The multiple curves correspond to random seeds of the original full-rank network, reproducing the results across a range of full-rank networks trained to do the same task. They show that different seeds can lead to solutions of different dimensionality (see for DMS for example one curve that stands out). This is now explained in the caption.
> - Equivalent of Fig 2f for the DMS task? Producing this figure is numerically particularly demanding, we can add it as a supplement in the final paper.

---

> > ### Comment · Reviewer_98wL · 2022-08-06
> > **Thanks for the rebuttal.**
> >
> > Thank you for the rebuttal. The K-back task is a nice addition, although I would argue that for the values of K explored (2-6) it is still low-dimensional.

---

### Author Response · Authors · 2022-08-01
**General rebuttal**

We thank all the reviewers for the time and effort invested in evaluating our work, and their encouraging comments. Here we address the comments common to several reviews, and summarize the corresponding changes to the paper. Individual responses to each reviewer are provided separately.

Novelty: This work builds on previous studies that fall into two classes: (i) theoretical analysis of low-rank RNNs trained or designed to perform specific tasks (Mastrogiuseppe & Ostojic 2018, Dubreuil et al 2022, Schuessler et al 2019 notably); (ii) data analyses of full-rank RNNs trained to fit recorded neural activity [Refs]. The key novelty of this work is that we combine these two approaches: we train low-rank RNNs to fit neural activity, and then use the low-rank theory to analyze and interpret the obtained networks. In the revised paper, this will be clarified in the introduction.
Methods for automatic discovery of interpretable models of neural activity are an important goal for neuroscience and ML research, and we believe our work provides a significant contribution to this literature.

Low-dimensionality of tasks: We focused on tasks used with animals in the neuroscience literature. These tasks are simple from an ML perspective, but challenging for animals to learn, and important for identifying mechanisms (for recent examples on the CDM task, see (Krause et al., biorXiv 2022), (Langdon & Engel, biorXiv 2022), (Flesch et al., Neuron 2022))
We nevertheless agree with reviewers that it is important to consider higher-dimensional tasks. During the revision period, we have added an analysis of the K-back task, in which a network receives a sequence of P stimuli in {-1, +1} (P being a random integer), and has to output the K-before-last stimulus received. This task requires the network to implement a working memory of capacity K, and the rank of a network implementing it should scale at least as K. We trained full-rank networks on this task for increasing values of K, and observed that their trajectories, through the LINT method, could indeed be fitted by low-rank networks of rank at least K (and usually very close to K). These results show that accommodating for increasingly long-term sequential or temporal effects requires an appropriate increase in dimensionality (as pointed notably by reviewer 8DBW citing (Cueva et al PNAS 2020)). Still, even confronted with higher-dimensional implementations, our method proves to be useful in quantifying precisely the dimensionality required by certain computations.
 We have uploaded a revised manuscript that includes a preliminary version of this analysis (see appendix F and Sup. Fig. 8 for details) and will add a more thorough version in the final revised manuscript.

Limitations of low-rank networks: The reviewers are right in pointing out that the limitations of our method are not sufficiently described. We wish to clarify them partially with the new results on the K-back task, and in the revised paper, we will rework the introduction and discussion to better clarify the limitations.


Applicability to other architectures: To which extent our method applies to other recurrent architectures like GRU and LSTM is a very interesting question that merits a study on its own. Recent analyses of dynamics and structure in those networks (see Maheswaranathan et al. ICML 2019, and NeurIPS 2019, or Schuessler et al. NeurIPS 2019 for example) suggest that low-rank models, even vanilla ones, could capture an important part of the variance of those trajectories. We will incorporate this point in the discussion, and aim to direct future research towards it.

We are also very sorry to discover that the pdf appeared as a series of images. This was unintentional, and caused by a technical issue. We have uploaded a proper pdf with searchable text now.

Tracked changes: removed wrap-ups of sections 3.1 and 3.3, added a paragraph l. 155-163, and another l. 296-300, added appendix F and SI Fig. 8, minor clarifications and typos.

---

### Meta-Review · Area_Chair_YBpJ · 2022-08-29

**Recommendation:** Accept
**Confidence:** Certain

**Metareview:**

Building on recent theoretical work on the dynamics of low-rank recurrent neural networks, the authors present a method called LINT for learning low-rank network models directly from data. As the reviewers point out, from a purely technical perspective, the idea is straightforward: simply optimize a low-rank parameterization of an RNN. Similar ideas have been considered under the heading of _tensorized_ neural networks [e.g. 1]. See also related works cited therein, which considers low-rank parameterizations of weight matrices [e.g. 2].

Though the technical innovation may be limited, the work makes up for it with connections to recent research in theoretical neuroscience and interesting experiments. The reviewers raise many important caveats and limitations (e.g. are these tasks really reflective of the complexity of "real world" tasks in ML and experimental neuroscience?). Overall, though, the reviewers and I think this paper offers valuable contributions. I encourage the authors to revise their manuscript in light of these thorough and constructive reviews.

[1] Novikov, A., Podoprikhin, D., Osokin, A. and Vetrov, D.P., 2015. Tensorizing neural networks. Advances in neural information processing systems, 28.

[2] Denil, M., Shakibi, B., Dinh, L., Ranzato, M.A. and De Freitas, N., 2013. Predicting parameters in deep learning. Advances in neural information processing systems, 26.

**Award:**

No

---

### Decision · Program_Chairs · 2022-09-14

Accept